# NeRS: Neural Reflectance Surfaces for Sparse-view 3D Reconstruction in the Wild

**Jason Y. Zhang**  **Gengshan Yang**  **Shubham Tulsiani**[*]  **Deva Ramanan**[*]

Robotics Institute, Carnegie Mellon University

## Abstract

Recent history has seen a tremendous growth of work exploring implicit representations of geometry and radiance, popularized through Neural Radiance Fields (NeRF). Such works are fundamentally based on a (implicit) *volumetric* representation of occupancy, allowing them to model diverse scene structure including translucent objects and atmospheric obscurants. But because the vast majority of real-world scenes are composed of well-defined surfaces, we introduce a *surface* analog of such implicit models called Neural Reflectance Surfaces (NeRS). NeRS learns a neural shape representation of a closed surface that is diffeomorphic to a sphere, guaranteeing water-tight reconstructions. Even more importantly, surface parameterizations allow NeRS to learn (neural) bidirectional surface reflectance functions (BRDFs) that factorize view-dependent appearance into environmental illumination, diffuse color (albedo), and specular "shininess." Finally, rather than illustrating our results on synthetic scenes or controlled in-the-lab capture, we assemble a novel dataset of multi-view images from online marketplaces for selling goods. Such "in-the-wild" multi-view image sets pose a number of challenges, including a small number of views with unknown/rough camera estimates. We demonstrate that surface-based neural reconstructions enable learning from such data, outperforming volumetric neural rendering-based reconstructions. We hope that NeRS serves as a first step toward building scalable, high-quality libraries of real-world shape, materials, and illumination. The project page with code and video visualizations can be found at jasonyzhang.com/ners.

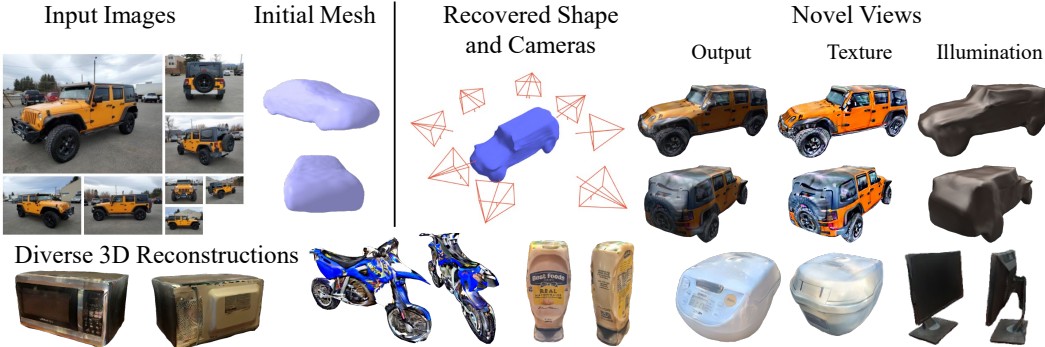

Figure 1: **3D view synthesis in the wild.** From several multi-view internet images of a truck and a coarse initial mesh (top left), we recover the camera poses, 3D shape, texture, and illumination (top right). We demonstrate the scalability of our approach on a wide variety of indoor and outdoor object categories (second row). [Video]

---

[*] denotes equal coding. Corresponding email: jasonyzhang@cmu.edu.

35th Conference on Neural Information Processing Systems (NeurIPS 2021).

# 1 Introduction

Although we observe the surrounding world only via 2D percepts, it is undeniably 3D. The goal of recovering this underlying 3D from 2D observations has been a longstanding one in the vision community, and any computational approach aimed at this task must answer a central question about representation—how should we model the geometry and appearance of the underlying 3D structure?

An increasingly popular answer to this question is to leverage neural *volumetric* representations of density and radiance fields (Mildenhall et al., 2020). This allows modeling structures from rigid objects to translucent fluids, while further enabling arbitrary view-dependent lighting effects. However, it is precisely this unconstrained expressivity that makes it less robust and unsuitable for modeling 3D objects from sparse views in the wild. While these neural volumetric representations have been incredibly successful, they require hundreds of images, typically with precise camera poses, to model the full 3D structure and appearance of real-world objects. In contrast, when applied to 'in-the-wild' settings *e.g.* a sparse set of images with imprecise camera estimates from off-the-shelf systems (see Fig. 1), they are unable to infer a coherent 3D representation. We argue this is because these neural volumetric representations, by allowing arbitrary densities and lighting, are *too* flexible.

Is there a robust alternative that captures real-world 3D structure? The vast majority of real-world objects and scenes comprise of well-defined *surfaces*. This implies that the geometry, rather than being an unconstrained volumetric function, can be modeled as a 2D manifold embedded in euclidean 3D space—and thus encoded via a (neural) mapping from a 2D manifold to 3D. Indeed, such meshed surface manifolds form the heart of virtually all rendering engines (Foley et al., 1996). Moreover, instead of allowing arbitrary view-dependent radiance, the appearance of such surfaces can be described using (neural) bidirectional surface reflection functions (BRDFs), themselves developed by the computer graphics community over decades. We operationalize these insights into *Neural Reflectance Surfaces* (NeRS), a surface-based neural representation for geometry and appearance.

NeRS represents shape using a neural displacement field over a canonical sphere, thus constraining the geometry to be a watertight surface. This representation crucially associates a surface normal to each point, which enables modeling view-dependent lighting effects in a physically grounded manner. Unlike volumetric representations which allow unconstrained radiance, NeRS factorizes surface appearance using a combination of diffuse color (albedo) and specularity. It does so by learning neural texture fields over the sphere to capture the albedo at each surface point, while additionally inferring an environment map and surface material properties. This combination of a surface constraint and a factored appearance allows NeRS to learn efficiently and robustly from a sparse set of images in the wild, while being able to capture varying geometry and complex view-dependent appearance.

Using only a coarse category-level template and approximate camera poses, NeRS can reconstruct instances from a diverse set of classes. Instead of evaluating in a synthetic setup, we introduce a dataset sourced from marketplace settings where multiple images of a varied set of real-world objects under challenging illumination are easily available. We show NeRS significantly outperforms neural volumetric or classic mesh-based approaches in this challenging setup, and as illustrated in Fig. 1, is able to accurately model the view-dependent appearance via its disentangled representation. Finally, as cameras recovered in the wild are only approximate, we propose a new evaluation protocol for *in-the-wild* novel view synthesis in which cameras can be refined during both training *and* evaluation. We hope that our approach and results highlight the several advantages that neural surface representations offer, and that our work serves as a stepping stone for future investigations.

# 2 Related Work

**Surface-based 3D Representations.** As they enable efficient representation and rendering, polygonal meshes are widely used in vision and graphics. In particular, morphable models (Blanz and Vetter, 1999) allow parametrizing shapes as deformations of a canonical template and can even be learned from category-level image collections (Cashman and Fitzgibbon, 2012; Kar et al., 2015). With the advances in differentiable rendering (Kato et al., 2018; Laine et al., 2020; Ravi et al., 2020), these have also been leveraged in learning based frameworks for shape prediction (Kanazawa et al., 2018; Gkioxari et al., 2019; Goel et al., 2020) and view synthesis (Riegler and Koltun, 2020). Whereas these approaches use an explicit discrete mesh, some recent methods have proposed using continuous

neural surface parametrization like ours to represent shape (Groueix et al., 2018) and texture (Tulsiani et al., 2020; Bhattad et al., 2021).

However, all of these works leverage such surface representations for (coarse) single-view 3D prediction given a category-level training dataset. In contrast, our aim is to infer such a representation given multiple images of a single instance, and without prior training. Closer to this goal of representing a single instance in detail, contemporary approaches have shown the benefits of using videos (Li et al., 2020; Yang et al., 2021) to recover detailed shapes, but our work tackles a more challenging setup where correspondence/flow across images is not easily available. In addition, while these prior approaches infer the surface texture, they do not enable the view-dependent appearance effects that our representation can model.

**Volumetric 3D and Radiance Fields.** Volumetric representations for 3D serve as a common, and arguably more flexible alternative to surface based representations, and have been very popular for classical multi-view reconstruction approaches (Furukawa and Hernández, 2015). These have since been incorporated in deep-learning frameworks for shape prediction (Girdhar et al., 2016; Choy et al., 2016) and differentiable rendering (Yan et al., 2016; Tulsiani et al., 2017). Although these initial approaches used discrete volumetric grids, their continuous neural function analogues have since been proposed to allow finer shape (Mescheder et al., 2019; Park et al., 2019) and texture modeling (Oechsle et al., 2019).

Whereas the above methods typically aimed for category-level shape representation, subsequent approaches have shown particularly impressive results when using these representations to model a single instance from images (Sitzmann et al., 2019a; Thies et al., 2019; Sitzmann et al., 2019b) – which is the goal of our work. More recently, by leveraging an implicit representation in the form of a Neural Radiance Field, Mildenhall et al. (2020) showed the ability to model complex geometries and illumination from images. There has since been a flurry of impressive work to further push the boundaries of these representations and allow modeling deformation (Pumarola et al., 2020; Park et al., 2021), lighting variation (Martin-Brualla et al., 2021), and similar to ours, leveraging insights from surface rendering to model radiance (Yariv et al., 2020; Boss et al., 2021; Oechsle et al., 2021; Srinivasan et al., 2021; Zhang et al., 2021b; Wang et al., 2021; Yariv et al., 2021). However, unlike our approach which can efficiently learn from a sparse set of images with coarse cameras, these approaches rely on a dense set of multi-view images with precise camera localization to recover a coherent 3D structure of the scene. DietNeRF (Jain et al., 2021) reduces the number of images but requires precise cameras and semantic supervision. BARF (Lin et al., 2021) relaxes the constraint of precise cameras while foregoing view-dependent appearance and requiring a dense set of images. Other approaches (Bi et al., 2020; Zhang et al., 2021a) that learn material properties from sparse views require specialized illumination rigs.

**Multi-view Datasets.** Many datasets study the longstanding problem of multi-view reconstruction and view synthesis. However, they are often captured in controlled setups, small in scale, and not diverse enough to capture the span of real world objects. Middlebury (Seitz et al., 2006) benchmarks multi-view reconstruction, containing two objects with nearly Lambertian surfaces. DTU (Aanæs et al., 2016) contains eighty objects with various materials but is still captured in a lab with controlled lighting. Freiburg cars (Sedaghat and Brox, 2015) captures 360 degree videos of fifty-two outdoor cars for multi-view reconstruction. ETH3D (Schöps et al., 2019) and Tanks and Temples (Knapitsch et al., 2017) contain both indoor and outdoor scenes but are small in scale. Perhaps most relevant are large-scale datasets of real-world objects such as Redwood (Choi et al., 2016) and Stanford Products (Oh Song et al., 2016), but the data is dominated by single-views or small baseline videos. In contrast, our Multi-view Marketplace Cars (MVMC) dataset contains thousands of multi-view captures of in-the-wild objects under various illumination conditions, making it suitable for studying and benchmarking algorithms for multi-view reconstruction, view synthesis, and inverse rendering.

## 3 Method

Given a sparse set of input images of an object under natural lighting conditions, we aim to model its shape and appearance. While recent neural volumetric approaches share a similar goal, they require a dense set of views with precise camera information. Instead, our approach relies only on approximate camera pose estimates and a coarse category-level shape template. Our key insight is that instead of allowing unconstrained densities popularly used for volumetric representations, we can enforce

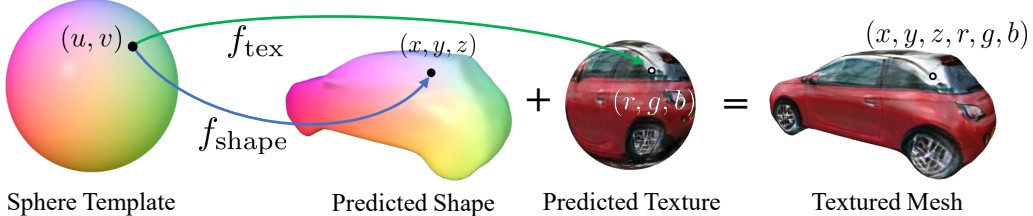

$(u, v)$  $f_{\text{tex}}$  $(x, y, z)$  $(x, y, z, r, g, b)$

$f_{\text{shape}}$  $+$  $(r, g, b)$  $=$

Sphere Template  Predicted Shape  Predicted Texture  Textured Mesh

Figure 2: **Neural Surface Representation.** We propose an implicit, continuous representation of shape and texture. We model shape as a deformation of a unit sphere via a neural network $f_{\text{shape}}$, and texture as a learned per-uv color value via a neural network $f_{\text{tex}}$. We can discretize $f_{\text{shape}}$ and $f_{\text{tex}}$ to produce the textured mesh above.

a *surface*-based 3D representation. Importantly, this allows view-dependent appearance variation by leveraging constrained reflection models that decompose appearance into diffuse and specular components. In this section, we first introduce our (neural) surface representation that captures the object's shape and texture, and then explain how illumination and specular effects can be modeled for rendering. Finally, we describe how our approach can learn using challenging in-the-wild images.

## 3.1 Neural Surface Representation

We represent object shape via a deformation of a unit sphere. Previous works (Kanazawa et al., 2018; Goel et al., 2020) have generally modeled such deformations *explicitly*: the unit sphere is discretized at some resolution as a 3D mesh with $V$ vertices. Predicting the shape deformation thus amounts to predicting vertex offsets $\delta \in \mathbb{R}^{V \times 3}$. Such *explicit discrete* representations have several drawbacks. First, they can be computationally expensive for dense meshes with fine details. Second, they lack useful spatial inductive biases as the vertex locations are predicted independently. Finally, the learned deformation model is fixed to a specific level of discretization, making it non-trivial, for instance, to allow for more resolution as needed in regions with richer detail. These limitations also extend to texture parametrization commonly used for such discrete mesh representations—using either per-vertex or per-face texture samples (Kato et al., 2018), or fixed resolution texture map, limits the ability to capture finer details.

Inspired by Groueix et al. (2018); Tulsiani et al. (2020), we address these challenges by adopting a continuous surface representation via a neural network. We illustrate this representation in Fig. 2. For any point $u$ on the surface of a unit sphere $\mathbb{S}^2$, we represent its 3D deformation $x \in \mathbb{R}^3$ using the mapping $f_{\text{shape}}(u) = x$ where $f_{\text{shape}}$ is parameterized as a multi-layer perceptron. This network therefore induces a deformation field over the surface of the unit sphere, and this deformed surface serves as our shape representation. We represent the surface texture in a similar manner–as a neural vector field over the surface of the sphere: $f_{\text{tex}}(u) = t \in \mathbb{R}^3$. This surface texture can be interpreted as an implicit UV texture map.

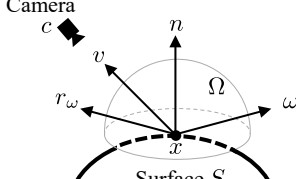

Figure 3: **Notation and convention for viewpoint and illumination parameterization.** The camera at $c$ is looking at point $x$ on the surface $S$. $v$ denotes the direction of the camera w.r.t $x$, and $n$ is the normal of $S$ at $x$. $\Omega$ denotes the unit hemisphere centered about $n$. We compute the light arriving in the direction of every $\omega \in \Omega$, and $r$ is the reflection of $w$ about $n$.

## 3.2 Modeling Illumination and Specular Rendering

**Surface Rendering.** The surface geometry and texture are not sufficient to infer appearance of the object *e.g.* a uniformly red car may appear darker on one side, and lighter on the other depending on the direction of incident light. In addition, depending on viewing direction and material properties, one may observe different appearance for the same 3D point *e.g.* shiny highlight from certain viewpoints. More formally, assuming that a surface does not emit light, the outgoing radiance $L_o$ in direction $v$ from a surface point $x$ can be described by the rendering equation (Kajiya, 1986; Immel et al., 1986):

$$L_o(x, v) = \int_{\Omega} f_r(x, v, \omega) L_i(x, \omega) (\omega \cdot n) d\omega \qquad (1)$$

where $\Omega$ is the unit hemisphere centered at surface normal $n$, and $\omega$ denotes the negative direction of incoming light. $f_r(x, v, \omega)$ is the bidirectional reflectance function (BRDF) which captures material

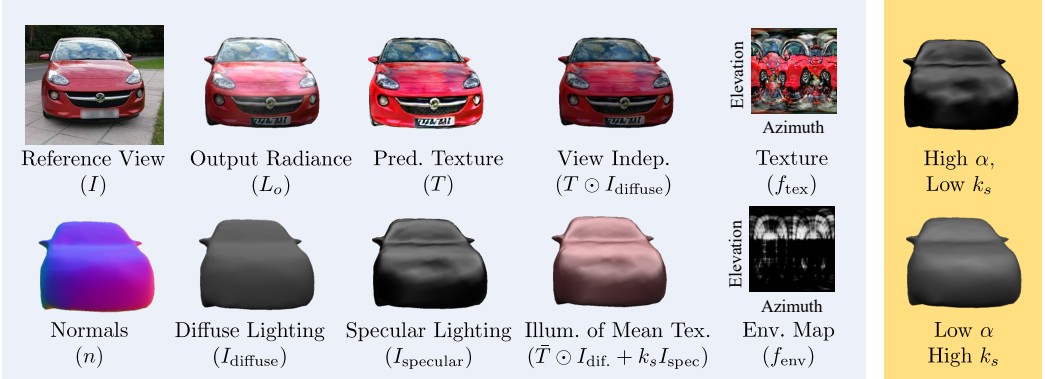

Figure 4: **Components of learned illumination model.** Given a query camera viewpoint (illustrated via the reference image $I$), we recover the radiance output $L_o$, computed using Phong shading (Phong, 1975). Here, we show the full decomposition of learned components. From the environment map $f_{env}$ and normals $n$, we compute diffuse ($I_{diffuse}$) and specular lighting ($I_{specular}$). The texture and diffuse lighting form the view-independent component ("View Indep.") and the specular lighting (weighted by the specular coefficient $k_s$) forms the view-dependent component of the radiance. Altogether, the output radiance $L_o = T \odot I_{diffuse} + k_s I_{specularity}$ (4). We also visualize the radiance using the mean texture, which is used to help learn plausible illumination. In the yellow box, we visualize the effects of the two specularity parameters. The shininess $\alpha$ controls the mirror-ness/roughness of the surface. The specular coefficient $k_s$ controls the intensity of the specular highlights.

properties (*e.g.* color and shininess) of surface $S$ at $x$, and $L_i(x, \omega)$ is the radiance coming toward $x$ from $\omega$ (Refer to Fig. 3). Intuitively, this integral computes the total effect of the reflection of every possible light ray $\omega$ hitting $x$ bouncing in the direction $v$.

We thus need to infer the environment lighting and surface material properties to allow realistic renderings. However, learning arbitrary lighting $L_i$ or reflection models $f_r$ is infeasible given sparse views, and we need to further constrain these to allow learning. Inspired by concurrent work (Wu et al., 2021) that demonstrated its efficacy when rendering rotationally symmetric objects, we leverage the Phong reflection model (Phong, 1975) with the lighting represented as a neural environment map.

**Neural Environment Map.** An environment map intuitively corresponds to the assumption that all the light sources are infinitely far away. This allows a simplified model of illumination, where the incoming radiance only depends on the direction $\omega$ and is independent of the position $x$ *i.e.* $L_i(x, \omega) \equiv I_\omega$. We implement this as a neural spherical environment map $f_{env}$ which learns to predict the incoming radiance for any query direction $L_i(x, \omega) \equiv I_\omega = f_{env}(\omega)$. Note that there is a fundamental ambiguity between material properties and illumination, *e.g.* a car that appears red could be a white car under red illumination, or a red car under white illumination. To avoid this, we follow Wu et al. (2021), and further constrain the environment illumination to be grayscale, *i.e.* $f_{env}(\omega) \in \mathbb{R}$.

**Appearance under Phong Reflection.** Instead of allowing an arbitrary BRDF $f_r$, the Phong reflection model decomposes the outgoing radiance from point $x$ in direction $v$ into the diffuse and specular components. The *view-independent* portion of the illumination is modeled by the diffuse component:

$$I_{diffuse}(x) = \sum_{\omega \in \Omega} (\omega \cdot n) I_\omega, \tag{2}$$

while the *view-dependent* portion of the illumination is modeled by the specular component:

$$I_{specular}(x, v) = \sum_{\omega \in \Omega} (r_{\omega,n} \cdot v)^\alpha I_\omega, \tag{3}$$

where $r_{\omega,n} = 2(\omega \cdot n)n - \omega$ is the reflection of $\omega$ about the normal $n$. The shininess coefficient $\alpha \in (0, \infty)$ is a property of the surface material and controls the "mirror-ness" of the surface. If $\alpha$ is high, the specular highlight will only be visible if $v$ aligns closely with $r_\omega$. Altogether, we compute the radiance of $x$ in direction $v$ as:

$$L_o(x, v) = T(x) \cdot I_{diffuse}(x) + k_s \cdot I_{specular}(x, v) \tag{4}$$

| Images | Init. Shape | Pred. Shape | Reconstructions from Novel Views |
|--------|-------------|-------------|----------------------------------|

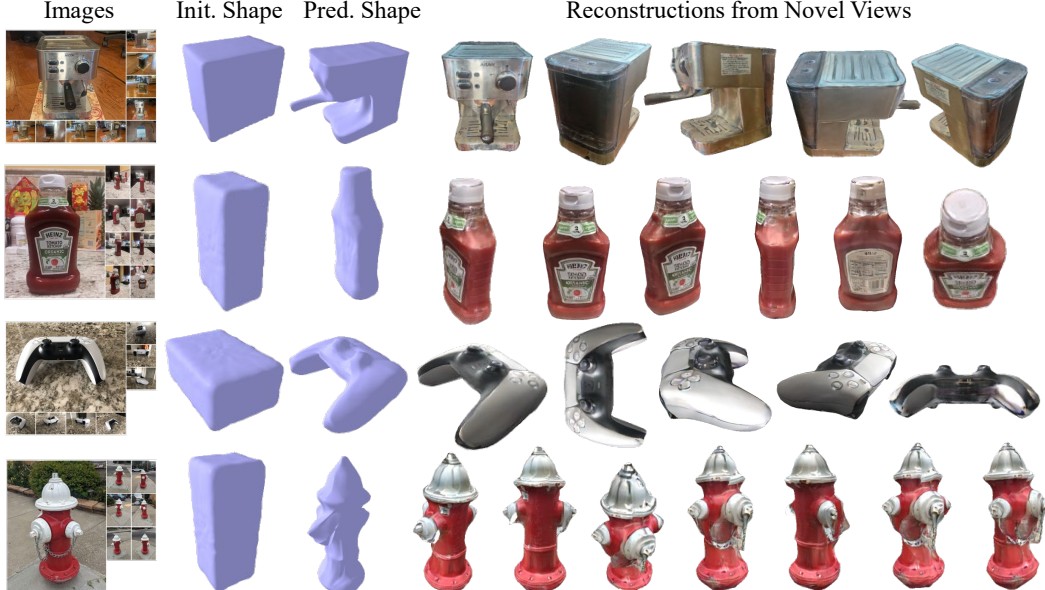

Figure 5: **Qualitative results on various household objects.** We demonstrate the versatility of our approach on an espresso machine, a bottle of ketchup, a game controller, and a fire hydrant. Each instance has 7-10 input views. We find that a coarse, cuboid mesh is sufficient as an initialization to learn detailed shape and texture. We initialize the camera poses by hand, roughly binning in increments of 45 degrees azimuth. [Video]

where the specularity coefficient $k_{\text{s}}$ is another surface material property that controls the intensity of the specular highlight. $T(x)$ is the texture value at $x$ computed by $f_{\text{tex}}$. For the sake of simplicity, $\alpha$ and $k_s$ are shared across the entire instance. See Fig. 4 for a full decomposition of these components.

## 3.3 Learning NeRS in the Wild

Given a sparse set of images in-the-wild, our approach aims to infer a NeRS representation, which when rendered, matches the available input. Concretely, our method takes as input $N$ (typically 8) images of the same instance $\{I_i\}_{i=1}^{N}$, noisy camera rotations $\{R_i\}_{i=1}^{N}$, and a category-specific mesh initialization $\mathcal{M}$. Using these, we aim to optimize full perspective cameras $\{\Pi\}_{i=1}^{N}$ as well as the neural surface shape $f_{\text{shape}}$, surface texture $f_{\text{text}}$, and environment map $f_{\text{env}}$. In addition, we also recover the material properties of the object, parametrized by a specularity coefficient $k_s$ and shininess coefficient $\alpha$.

**Initialization.** Note that both the camera poses and mesh initialization are only required to be coarsely accurate. We use an off-the-shelf approach (Xiao et al., 2019) to predict camera rotations, and we find that a cuboid is sufficient as an initialization for several instances (See Fig. 5). We use off-the-shelf approaches (Rother et al., 2004; Kirillov et al., 2020) to compute masks $\{M_i\}_{i=1}^{N}$. We assume that all images were taken with the same camera intrinsics. We initialize the shared global focal length $f$ to correspond to a field of view of 60 degrees, and set the principal point at center of each image. We initialize the camera pose with the noisy initial rotations $R_i$ and a translation $t_i$ such that the object is fully in view. We pre-train $f_{\text{shape}}$ to output the template mesh $\mathcal{M}$.

**Rendering.** To render an image, NeRS first discretizes the neural shape model $f_{\text{shape}}(u)$ over spherical coordinates $u$ to construct an explicit triangulated surface mesh. This triangulated mesh and camera $\Pi_i$ are fed into PyTorch3D's differentiable renderer (Ravi et al., 2020) to obtain per-pixel (continuous) spherical coordinates and associated surface properties:

$$[UV, N, \hat{M}_i] = \text{Rasterize}(\pi_i, f_{\text{shape}}) \tag{5}$$

where $UV[p]$, $N[p]$, and $\hat{M}[p]$ are (spherical) uv-coordinates, normals, and binary foreground-background labels corresponding to each image pixel $p$. Together with the environment map $f_{\text{env}}$ and specular material parameters $(\alpha, k_s)$, these quantities are sufficient to compute the outgoing radiance at each pixel $p$ under camera viewpoint $\Pi_i$ using (4). In particular, denoting by $v(\Pi, p)$ the viewing

direction for pixel $p$ under camera $\Pi$, and using $u \equiv UV[p], n \equiv N[p]$ for notational brevity, the intensity at pixel $p$ can be computed as:

$$\hat{I}[p] = f_{\text{tex}}(u) \cdot \Big( \sum_{\omega \in \Omega} (\omega \cdot n) \, f_{\text{env}}(\omega) \Big) + k_s \Big( \sum_{\omega \in \Omega} (r_{\omega,n} \cdot v(\Pi, p))^a \, f_{\text{env}}(\omega) \Big) \tag{6}$$

**Image loss.** We compute a perceptual loss (Zhang et al., 2018) $L_{\text{perceptual}}(I_i, \hat{I}_i)$ that compares the distance between the rendered and true image using off-the-shelf VGG deep features. Note that being able to compute a perceptual loss is a significant benefit of surface-based representations over volumetric approaches such as NeRF (Mildenhall et al., 2020), which operate on batches of rays rather than images, due to the computational cost of volumetric rendering. Similar to Wu et al. (2021), we find that an additional rendering loss using the mean texture (see Fig. 4 and Fig. 8 for examples with details in the supplement) helps learn visually plausible lighting.

**Mask Loss.** To measure disagreement between the rendered and measured silhouettes, we compute a mask loss $L_{\text{mask}} = \frac{1}{N} \sum_{i=1}^{N} \|M_i - \hat{M}_i\|_2^2$, distance transform loss $L_{\text{dt}} = \frac{1}{N} \sum_{i=1}^{N} D_i \odot \hat{M}_i$, and 2D chamfer loss $L_{\text{chamfer}} = \frac{1}{N} \sum_{i=1}^{N} \sum_{p \in E(M_i)} \min_{\hat{p} \in \hat{M}_i} \|p - \hat{p}\|_2^2$. $D_i$ refers to the Euclidean distance transform of mask $M_i$, $E(\cdot)$ computes the 2D pixel coordinates of the edge of a mask, and $\hat{p}$ is every pixel coordinate in the predicted silhouette.

**Regularization.** Finally, to encourage smoother shape whenever possible, we incorporate a mesh regularization loss $L_{\text{regularize}} = L_{\text{normals}} + L_{\text{laplacian}}$ consisting of normals consistency and Laplacian smoothing losses (Nealen et al., 2006; Desbrun et al., 1999). Note that such geometry regularization is another benefit of surface representations over volumetric ones. Altogether, we minimize:

$$L = \lambda_1 L_{\text{mask}} + \lambda_2 L_{\text{dt}} + \lambda_3 L_{\text{chamfer}} + \lambda_4 L_{\text{perceptual}} + \lambda_5 L_{\text{regularize}} \tag{7}$$

w.r.t $\Pi_i = [R_i, t_i, f], \alpha, k_s$, and the weights of $f_{\text{shape}}$, $f_{\text{text}}$, and $f_{\text{env\_map}}$.

**Optimization.** We optimize (7) in a coarse-to-fine fashion, starting with a few parameters and slowly increasing the number of free parameters. We initially optimize (7), w.r.t only the camera parameters $\Pi_i$. After convergence, we sequentially optimize $f_{\text{shape}}$, $f_{\text{tex}}$, and $f_{\text{env}}/\alpha/k_s$. We find it helpful to sample a new set of spherical coordinates $u$ each iteration when rasterizing. This helps propagate gradients over a larger surface and prevent aliasing. With 4 Nvidia 1080TI GPUs, training NeRS requires approximately 30 minutes. Please see Sec. **??** for hyperparameters and additional details.

# 4 Evaluation

In this section, we demonstrate the versatility of Neural Reflectance Surfaces to recover meaningful shape, texture, and illumination from in-the-wild indoor and outdoor images.

**Multi-view Marketplace Dataset.** To address the shortage of in-the-wild multi-view datasets, we introduce a new dataset, Multi-view Marketplace Cars (MVMC), collected from an online marketplace with thousands of car listings. Each user-submitted listing contains seller images of the same car instance. In total, we curate a subset of size 600 with at least 8 exterior views (averaging 10 exterior images per listing) along with 20 instances for an evaluation set (averaging 9.1 images per listing). We use Xiao et al. (2019) to compute rough camera poses. MVMC contains a large variety of cars under various illumination conditions (*e.g.* indoors, overcast, sunny, snowy, etc). The filtered dataset with anonymized personally identifiable information (*e.g.* license plates and phone numbers), masks, initial camera poses, and optimized NeRS cameras will be made available on the project page.

**Novel View Synthesis.** Traditionally, novel view synthesis requires accurate target cameras to use as queries. Existing approaches use COLMAP (Schönberger and Frahm, 2016) to recover ground truth cameras, but this consistently fails on MVMC due to specularities and limited views. On the other hand, we can use learning-based methods (Xiao et al., 2019) to recover camera poses for both training and test views. However, as these are inherently approximate, this complicates training *and* evaluation. To account for this, we explore two evaluation protocols. First, to mimic the traditional evaluation setup, we obtain pseudo-ground truth cameras (with manual correction) and freeze them during training and evaluation. While this evaluates the quality of the 3D reconstruction, it does not evaluate the method's ability to jointly recover cameras. As a more realistic setup for evaluating view synthesis in the wild, we evaluate each method with approximate (off-the-shelf) cameras, while allowing them to be optimized.

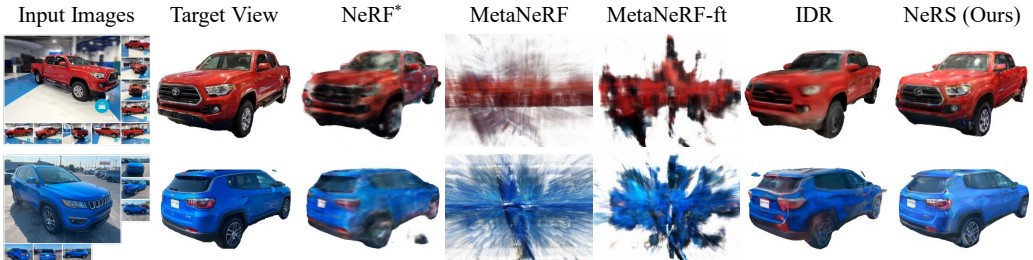

| Input Images | Target View | NeRF* | MetaNeRF | MetaNeRF-ft | IDR | NeRS (Ours) |

Figure 6: **Qualitative comparison *with fixed cameras*.** We evaluate all baselines on the task of novel view synthesis on Multi-view Marketplace Cars trained and tested with fixed, pseudo-ground truth cameras. One image is held out during training. Since we do not have ground truth cameras, we treat the optimized cameras from optimizing over all images as the ground truth cameras. We train a modified version (See Sec. 4) of NeRF (Mildenhall et al., 2020) that is more competitive with sparse views (NeRF*). We also evaluate against a meta-learned initialization of NeRF with and without finetuning until convergence (Tancik et al., 2021), but found poor results perhaps due to the domain shift from Shapenet cars. Finally, IDR (Yariv et al., 2020) extracts a surface from an SDF representation, but struggles to produce a view-consistent output given limited input views. We find that NeRS synthesizes novel views that are qualitatively closer to the target. The red truck has 16 total views while the blue SUV has 8 total views. [Video]

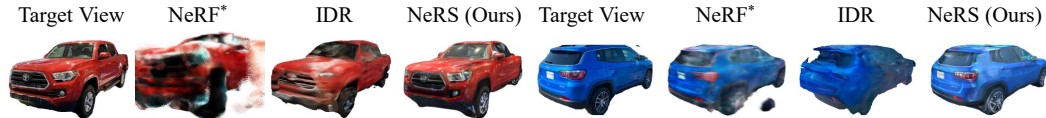

| Target View | NeRF* | IDR | NeRS (Ours) | Target View | NeRF* | IDR | NeRS (Ours) |

Figure 7: **Qualitative results for *in-the-wild* novel view synthesis.** Since off-the-shelf camera poses are only approximate for both training and test images, we allow cameras to be optimized during both training and evaluation (See Tab. 2 and Sec. 4). We find that NeRS generalizes better than the baselines in this unconstrained but more realistic setup. [Video]

**Novel View Synthesis *with Fixed Cameras*.** In the absence of ground truth cameras, we create pseudo-ground truth by manually correcting cameras recovered by jointly optimizing over all images for each object instance. For each evaluation, we treat one image-camera pair as the target and the remaining pairs for training. We repeat this process for each image in the evaluation set (totaling 182). Unless otherwise noted, qualitative results use approximate cameras and not the pseudo-ground truth.

**Novel View Synthesis *in the Wild*.** While the above evaluates the quality of the 3D reconstructions, it is not representative of in-the-wild settings where the initial cameras are unknown/approximate and should be optimized during training. Because even the test camera is approximate, each method is similarly allowed to refine the test camera to better match the test image while keeping the model fixed. Intuitively, this measures the ability of a model to synthesize a target view under *some* camera. See implementation details in the supplement.

**Baselines.** We evaluate our approach against Neural Radiance Fields (NeRF) (Mildenhall et al., 2020), which learns a radiance field conditioned on viewpoint and position and renders images using raymarching. We find that the vanilla NeRF struggles in our in-the-wild low-data regime. As such, we make a number of changes to make the NeRF baseline (denoted NeRF*) as competitive as possible, including a mask loss and a canonical volume. Please see the appendix for full details. We also evaluate a simplified NeRF with a meta-learned initialization for cars from multi-view images (Tancik et al., 2021), denoted as MetaNeRF. MetaNeRF meta-learns an initialization such that with just a few gradient steps, it can learn a NeRF model. This allows the model to learn a data-driven prior over the shape of cars. Note that MetaNeRF is trained on ShapeNet (Chang et al., 2015) and thus has seen more data than the other test-time-optimization approaches. We find that the default number of gradient steps was insufficient for MetaNeRF to converge on images from MVMC, so we also evaluate MetaNeRF-ft, which is finetuned until convergence. Finally, we evaluate IDR (Yariv et al., 2020), which represents geometry by extracting a surface from a signed distance field. IDR learns a neural renderer conditioned on the camera direction, position, and normal of the surface.

**Metrics.** We evaluate all approaches using the traditional image similarity metrics Mean-Square Error (MSE), Peak Signal-to-Noise Ratio (PSNR), and Structural Similarity Index Measure (SSIM).

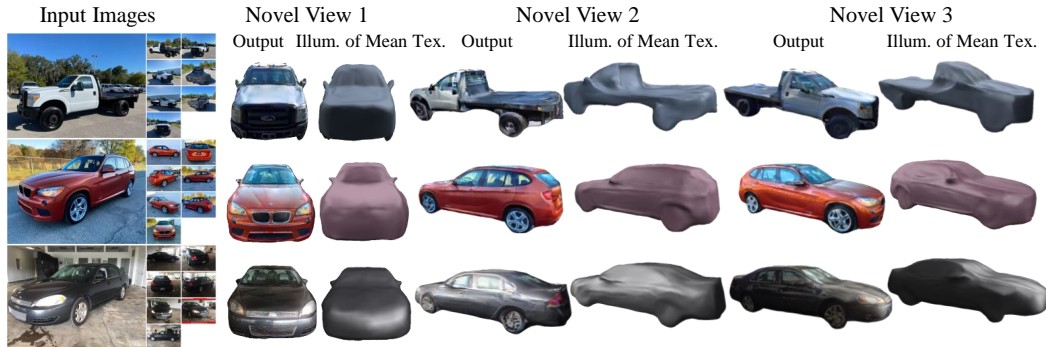

Figure 8: **Qualitative results on our in-the-wild Multi-view Marketplace Cars dataset.** Here we visualize the NeRS outputs as well as the illumination of the mean texture on 3 of listings from the MVMC dataset. We find that NeRS recovers detailed textures and plausible illumination. Each instance has 8 input views. [Video]

| Method | MSE ↓ | PSNR ↑ | SSIM ↑ | LPIPS ↓ | FID ↓ |
|---|---|---|---|---|---|
| NeRF* (Mildenhall et al., 2020) | 0.0393 | 16.0 | 0.698 | 0.287 | 231.7 |
| MetaNeRF (Tancik et al., 2021) | 0.0755 | 11.4 | 0.345 | 0.666 | 394.5 |
| MetaNeRF-ft (Tancik et al., 2021) | 0.0791 | 11.3 | 0.500 | 0.542 | 326.8 |
| IDR (Yariv et al., 2020) | 0.0698 | 13.8 | 0.658 | 0.328 | 190.1 |
| NeRS (Ours) | **0.0254** | **16.5** | **0.720** | **0.172** | **60.9** |

Table 1: **Quantitative evaluation of novel-view synthesis on MVMC using *fixed pseudo-ground truth cameras*.** To evaluate novel view synthesis in a manner consistent with previous works that assume known cameras, we obtain pseudo-ground truth cameras by manually correcting off-the-shelf recovered cameras. We evaluate against a modified NeRF (NeRF*), a meta-learned initialization to NeRF with and without finetuning (MetaN-eRF), and the volumetric surface-based IDR. NeRS significantly outperforms the baselines on all metrics on the task of novel-view synthesis with fixed cameras. See Fig. 6 for qualitative results.

| Method | MSE ↓ | PSNR ↑ | SSIM ↑ | LPIPS ↓ | FID ↓ |
|---|---|---|---|---|---|
| NeRF* (Mildenhall et al., 2020) | 0.0464 | 14.7 | 0.660 | 0.335 | 277.9 |
| IDR (Yariv et al., 2020) | 0.0454 | 14.4 | 0.685 | 0.297 | 242.3 |
| NeRS (Ours) | **0.0338** | **15.4** | 0.675 | **0.221** | **92.5** |

Table 2: **Quantitative evaluation of *in-the-wild* novel-view synthesis on MVMC.** Off-the-shelf cameras estimated for in-the-wild data are inherently erroneous. This means that both training and test cameras are approximate, complicating training *and* evaluation. To compensate for approximate test cameras, we allow methods to refine the test camera given the test image with the model fixed (see Sec. **??** for details). Intuitively this measures the ability of a method to synthesize a test image under *some* camera. We evaluate against NeRF and IDR, and find that NeRS outperforms the baselines across all metrics. See Fig. 7 for qualitative results.

We also compute the Learned Perceptual Image Patch Similarity (LPIPS) (Zhang et al., 2018) which correlates more strongly with human perceptual distance. Finally, we compute the Fréchet Inception Distance (Heusel et al., 2017) between the novel view renderings and original images as a measure of visual realism. In Tab. 1 and Tab. 2, we find that NeRS significantly outperforms the baselines in all metrics across both the fixed camera and in-the-wild novel-view synthesis evaluations. See Fig. 6 and Fig. 7 for a visual comparison of the methods.

**Qualitative Results.** In Fig. 8, we show qualitative results on our Multi-view Marketplace Cars dataset. Each car instance has between 8 and 16 views. We visualize the outputs of our reconstruction from 3 novel views. We show the rendering for both the full radiance model and the mean texture. Both of these renderings are used to compute the perceptual loss (See Sec. 3.2). We find that NeRS recovers detailed texture information and plausible illumination parameters. To demonstrate the scalability of our approach, we also evaluate on various household objects in Fig. 5. We find that a coarse, cuboid mesh is sufficient as an initialization to recover detailed shape, texture, and lighting conditions. Please refer to the project webpage for 360 degree visualizations.

# 5 Discussion

We present NeRS, an approach for learning neural surface models that capture geometry and surface reflectance. In contrast to volumetric neural rendering, NeRS enforces watertight and closed manifolds. This allows NeRS to model surface-based appearance affects, including view-dependant specularities and normal-dependant diffuse appearance. We demonstrate that such regularized reconstructions allow for learning from sparse in-the-wild multi-view data, enabling reconstruction of objects with diverse material properties across a variety of indoor/outdoor illumination conditions. Further, the recovery of accurate camera poses in the wild (where classic structure-from-motion fails) remains unsolved and serves as a significant bottleneck for all approaches, including ours. We tackle this problem by using realistic but approximate off-the-shelf camera poses and by introducing a new evaluation protocol that accounts for this. We hope NeRS inspires future work that evaluates *in the wild* and enables the construction of high-quality libraries of real-world geometry, materials, and environments through better neural approximations of shape, reflectance, and illuminants.

**Limitations.** Though NeRS makes use of factorized models of illumination and material reflectance, there exists some fundamental ambiguities that are difficult from which to recover. For example, it is difficult to distinguish between an image of a gray car under bright illumination and an image of a white car under dark illumination. We visualize such limitations in the supplement. In addition, because the neural shape representation of NeRS is diffeomorphic to a sphere, it cannot model objects of non-genus-zero topologies.

**Broader Impacts.** NeRS reconstructions could be used to reveal identifiable or proprietary information (e.g., license plates). We made our best effort to blur out all license plates and other personally-identifiable information in the paper and data.

**Acknowledgements.** We would like to thank Angjoo Kanazawa and Leonid Keselman for useful discussion and Sudeep Dasari and Helen Jiang for helpful feedback on drafts of the manuscript. This work was supported in part by the NSF GFRP (Grant No. DGE1745016), Singapore DSTA, and CMU Argo AI Center for Autonomous Vehicle Research.

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
