# Neural Reflectance Surfaces (NeRS)
## *Supplementary Materials*

We describe an additional ablation on view-dependence in Sec. 0.1, some limitations of our illumination model in Sec. 0.2, implementation details for NeRS and the baselines in Sec. 0.3 and Sec. 0.4 respectively, and details for in-the-wild novel view synthesis in Sec. 0.5. We also evaluate the effect of increasing the number of training images on novel view synthesis in Fig. 1, ablate removing any view-dependence prediction altogether in Fig. 2, compare the learned shape models with volume carving in Fig. 4, and describe the architectures of NeRS networks in Fig. 3.

## 0.1 View-Dependence Ablation

To evaluate the importance of learning a BRDF for illumination estimation, we train a NeRS that directly conditions the radiance prediction on the position and view direction, similar to NeRF. More specifically, we concatenate the $uv$ with view direction $\omega$ as the input to $f_{\text{tex}}$ which still predicts an RGB color value, and do not use $f_{\text{env}}$. We include the quantitative evaluation on using our Multi-view Marketplace Cars (MVMC) dataset for fixed camera novel-view synthesis in Tab. 1 and *in-the-wild* novel-view synthesis in Tab. 2.

Video qualitative results are available on the figures page[1] of the project webpage. We observe that the NeRS with NeRF-style view-direction conditioning has qualitatively similar visual artifacts to the NeRF baseline, particularly large changes in appearance despite small changes in viewing direction. This suggests that learning a BRDF is an effective way to regularize and improve generalization of view-dependent effects.

| Method | MSE ↓ | PSNR ↑ | SSIM ↑ | LPIPS ↓ | FID ↓ |
|---|---|---|---|---|---|
| NeRF* (Mildenhall et al., 2020) | 0.0393 | 16.0 | 0.698 | 0.287 | 231.7 |
| MetaNeRF (Tancik et al., 2021) | 0.0755 | 11.4 | 0.345 | 0.666 | 394.5 |
| MetaNeRF-ft (Tancik et al., 2021) | 0.0791 | 11.3 | 0.500 | 0.542 | 326.8 |
| IDR (Yariv et al., 2020) | 0.0698 | 13.8 | 0.658 | 0.328 | 190.1 |
| NeRS (Ours) | **0.0254** | **16.5** | **0.720** | **0.172** | **60.9** |
| NeRS + NeRF-style View-dep | 0.0315 | 15.6 | 0.68 | 0.271 | 285.3 |

Table 1: **Quantitative evaluation of novel-view synthesis on MVMC using *fixed pseudo-ground truth cameras*.** Here, we evaluate novel view synthesis with fixed pseudo-ground truth cameras, constructed by manually correcting off-the-shelf cameras that are jointly optimized by our method. In addition to the baselines from the main paper, we compare against an ablation of our approach that directly conditions the radiance on the $uv$ and viewing direction ("NeRS + NeRF View-dep") in a manner similar to NeRF.

## 0.2 Limitations of illumination model

In order to regularize lighting effects, we assume all lighting predicted by $f_{\text{env}}$ to be grayscale. For some images with non-white lighting (see Fig. 5), this can cause the lighting color to be baked into the predicted texture in an unrealistic manner. In addition, if objects are gray in color, there is a fundamental ambiguity as to whether the luminance is due to the object texture or lighting *e.g.* dark car with bright illumination or light car with dark illumination (see Fig. 6).

---

[1] https://jasonyzhang.com/ners/paper_figures

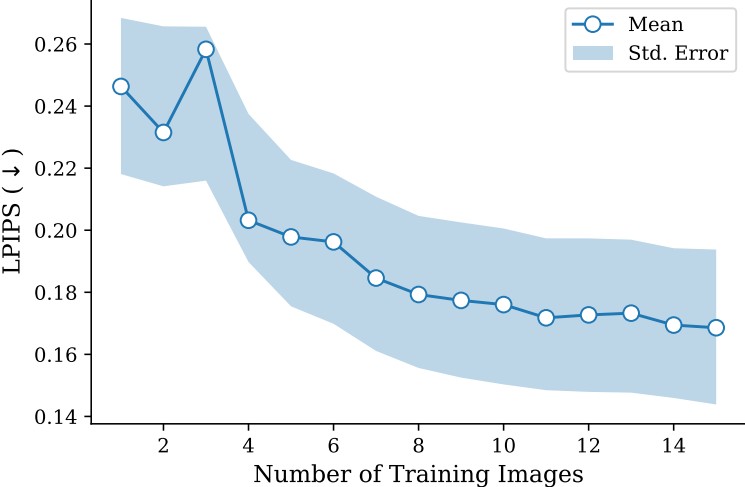

Figure 1: **Relationship between number of training images and reconstruction quality.** We quantify the number of images needed for a meaningful reconstruction using NeRS on a specific instance from MVMC with 16 total images. Given one of the images as a target image, we randomly select one of the remaining images as the initial training image. Then, we iteratively increase the number of training images, adding the image corresponding to the pseudo-ground truth pose furthest from the current set. This setup is intended to emulate how a user would actually take pictures of objects in the wild (*i.e.* taking wide baseline multi-view images from far away viewpoints). Note that once the sets of training images are selected, we use the *in-the-wild* novel view synthesis training and evaluation protocol. In this plot, we visualize the mean and standard error over 16 runs. We find that increasing the number of images improves the quality in terms of perceptual similarity, with performance beginning to plateau after 8 images.

| Target Camera | Method | MSE ↓ | PSNR ↑ | SSIM ↑ | LPIPS ↓ | FID ↓ |
|---|---|---|---|---|---|---|
| Approx. Camera | NeRF* (Mildenhall et al., 2020) | 0.0657 | 12.7 | 0.604 | 0.386 | 290.7 |
| | IDR (Yariv et al., 2020) | 0.0836 | 11.4 | 0.591 | 0.413 | 251.6 |
| | NeRS (Ours) | 0.0527 | 13.3 | 0.620 | 0.284 | 100.9 |
| | NeRS + NeRF-style View-dep. | 0.0573 | 13.0 | 0.624 | 0.356 | 296.4 |
| Refined Approx. Camera | NeRF* (Mildenhall et al., 2020) | 0.0464 | 14.7 | 0.660 | 0.335 | 277.9 |
| | IDR (Yariv et al., 2020) | 0.0454 | 14.4 | 0.685 | 0.297 | 242.3 |
| | NeRS (Ours) | **0.0338** | **15.4** | **0.675** | **0.221** | **92.5** |
| | NeRS + NeRF-style View-dep. | 0.0482 | 13.9 | 0.659 | 0.316 | 293.5 |

Table 2: **Quantitative evaluation of *in-the-wild* novel-view synthesis on MVMC.** To evaluate *in-the-wild* novel view synthesis, each method is trained with (and can refine) approximate cameras estimated using Xiao et al. (2019). To compensate for the approximate test camera, we allow each method to refine the test camera given the target image. In this table, we show the performance before ("Approx. Camera") and after ("Refined Approx. Camera") this refinement. All methods improve from the refinement. In addition, we compare against an ablation of our approach that directly conditions the radiance on the $uv$ and viewing direction ("NeRS + NeRF View-dep") in a manner similar to NeRF.

## 0.3    Implementation Details and Hyper-Parameters

To encourage plausible lighting, we found it helpful to also optimize a perceptual loss on the illumination of the mean texture, similar to Wu et al. (2021). During optimization, we render an image $\hat{I}$ with full texture using $f_{tex}$ as well as a rendering $\bar{I}$ using the mean output of $f_{tex}$. We compute a perceptual loss on both $\hat{I}$ and $\bar{I}$. We weight these two perceptual losses differently, as shown in Tab. 3. To compute illumination, we sample environment rays uniformly across the unit sphere, and compute the normals corresponding to each vertex, ignoring rays pointing in the opposite direction of the normal.

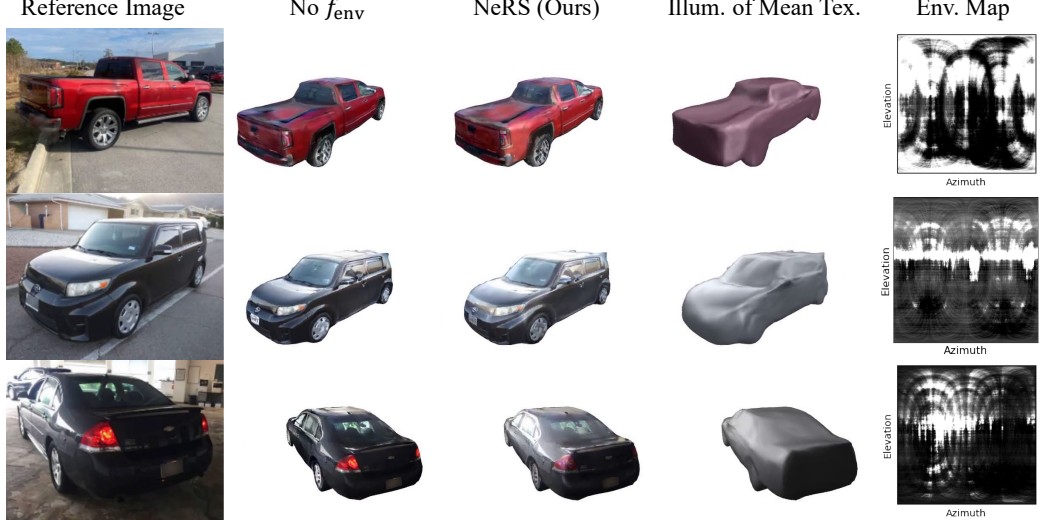

Figure 2: **Comparison with NeRS trained without view dependence.** Here we compare the full NeRS (column 3) with a NeRS trained without view dependence by only rendering using $f_{tex}$ without $f_{env}$ (column 2). We find that NeRS trained without view-dependence cannot capture lighting effects when they are inconsistent across images. The difference between NeRS trained with and without view-dependence is best viewed in video form. We also visualize the environment map and the illumination of the car with the mean texture. The environment maps show that the light is coming primarily from one side for the first car, uniformly from all directions for the second car, and strongly front left for the third car. [Video]

## 0.4    Implementation Details of Baselines

**NeRF**[2] (Mildenhall et al., 2020): We find that a vanilla NeRF struggles when given sparse views. As such, we make the following changes to make the NeRF baseline as competitive as possible: 1. we add a mask loss that forces rays to either pass through or be absorbed entirely by the neural volume, analogous to space carving (Kutulakos and Seitz, 2000) (see Fig. 4); 2. a canonical volume that zeros out the density outside of a tight box where the car is likely to be. This helps avoid spurious "cloudy" artifacts from novel views; 3. a smaller architecture to reduce overfitting; and 4. a single rendering pass rather than dual coarse/fine rendering passes. In the main paper, we denote this modified NeRF as NeRF*. For in-the-wild novel-view synthesis, we refine the training and test cameras directly, similar to Wang et al. (2021).

**MetaNeRF**[3] (Tancik et al., 2021): We fine-tuned the pre-trained 25-view ShapeNet model on our dataset. We used the default hyper-parameters for the MetaNeRF baseline. For MetaNeRF-ft, we increased the number of samples to 1024, trained with a learning rate of 0.5 for 20,000 iterations, then lowered the learning rate to 0.05 for 50,000 iterations.

**IDR**[4] (Yariv et al., 2020): Because each instance in our dataset has fewer images than DTU, we increased the number of epochs by 5 times and adjusted the learning rate scheduler accordingly. IDR supports both fixed cameras (which we use for *fixed* camera novel-view synthesis) and trained cameras (which we use for *in-the-wild* novel-view synthesis).

## 0.5    *In-the-wild* Novel-view Synthesis Details

For *in-the-wild* novel-view synthesis without ground truth cameras, we aim to evaluate the capability of each approach to recover a meaningful 3D representation while only using approximate off-the-shelf cameras. Given training images with associated approximate cameras, each method is required to output a 3D model. We then evaluate whether this 3D model can generate a held-out target image under *some* camera pose.

---

[2]https://github.com/facebookresearch/pytorch3d/tree/main/projects/nerf
[3]https://github.com/tancik/learnit
[4]https://github.com/lioryariv/idr

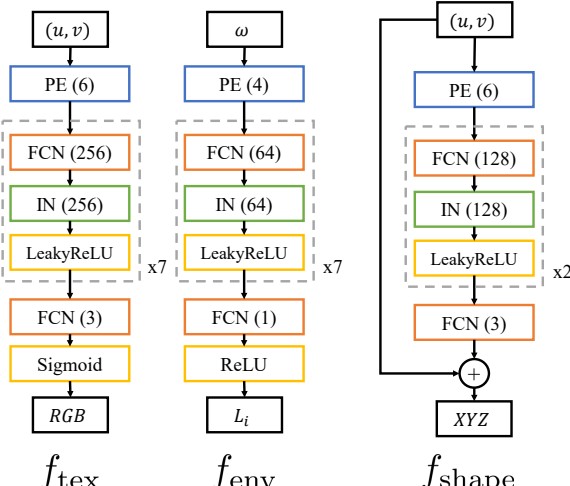

Figure 3: **Network Architectures.** Here, we show the architecture diagrams for $f_{\text{tex}}$, $f_{\text{env}}$, and $f_{\text{shape}}$. We parameterize $(u, v)$ coordinates as 3-D coordinates on the unit sphere. Following Mildenhall et al. (2020); Tancik et al. (2020), we use positional encoding to facilitate learning high frequency functions, with 6 sine and cosine bases (thus mapping 3-D to 36-D). We use blocks of fully connected layers, instance norm (Ulyanov et al., 2016), and Leaky ReLU. To ensure texture is between 0 and 1 and illumination is non-negative, we use a final sigmoid and ReLU activation for $f_{\text{tex}}$ and $f_{\text{env}}$ respectively. We pre-train $f_{\text{shape}}$ to output the category-specific mesh template. Given a category-specific mesh, we use an off-the-shelf approach to recover the mesh-to-sphere mapping: https://github.com/icemiliang/spherical_harmonic_maps.

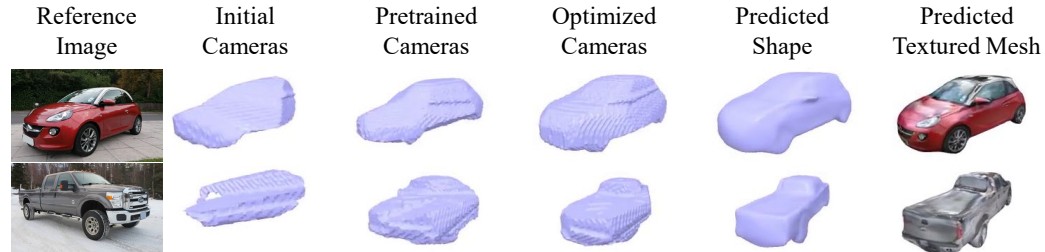

| Reference Image | Initial Cameras | Pretrained Cameras | Optimized Cameras | Predicted Shape | Predicted Textured Mesh |

Figure 4: **Shape from Silhouettes using Volume Carving.** We compare shapes carved from the silhouettes of the training views with the shape model learned by our approach. We construct a voxel grid of size $128^3$ and keep only the voxels that are visible when projected to the masks using the off-the-shelf cameras ("Initial Cameras"), pre-trained cameras from Stage 1 ("Pretrained Cameras"), and the final cameras after Stage 4 ("Optimized Cameras"). We compare this with the shape model output by $f_{\text{shape}}$. We show the nearest neighbor training view and the final NeRS rendering for reference. While volume carving can appear reasonable given sufficiently accurate cameras, we find that the shape model learned by NeRS is qualitatively a better reconstruction. In particular, the model learns to correctly output the right side of the pickup truck and reconstructs the sideview mirrors from the texture cues, suggesting that a joint optimization of the shape and appearance is useful. Also, we note that the more "accurate" optimized cameras are themselves outputs of NeRS. [Video]

In practice, we recover approximate cameras using Xiao et al. (2019). However, as these only comprise of a rotation, we further refine them using the initial template and foreground mask. The initial approximate cameras we use across all methods are in fact equivalent to those obtained by our method after Stage 1 of training, and depend upon: i) prediction from Xiao et al. (2019), and ii) input mask and template shape. During training, each method learns a 3D model while jointly optimizing the camera parameters using gradient descent. Similarly, at test time, we refine approximate test camera with respect to the training loss function for each method (*i.e.* (7) for NeRS, MSE for NeRF, and RGB+Mask loss for IDR) using the Adam optimizer for 400 steps. We show all metrics before and after this camera refinement in Tab. 2.

## References

Diederik P Kingma and Jimmy Ba. Adam: A Method for Stochastic Optimization. In *ICLR*, 2015.

| Reference Image | Output Radiance | Illum. of Mean Texture | Predicted Texture |

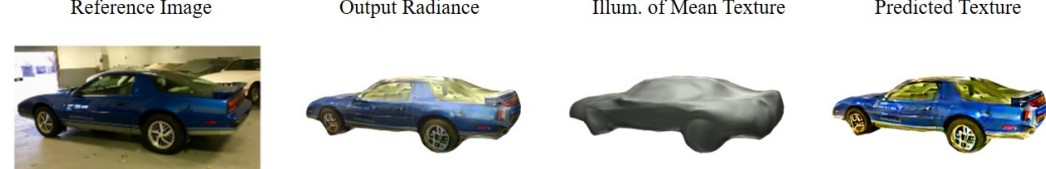

Figure 5: **Non-white lighting.** Our illumination model assumes environment lighting to be grayscale. In this car, the indoor lighting is yellow, which causes the yellow lighting hue to be baked into the predicted texture (Predicted Texture) rather than the illumination (Illum. of Mean Texture).

| Reference Image | Output Radiance | Illum. of Mean Texture | Predicted Texture |

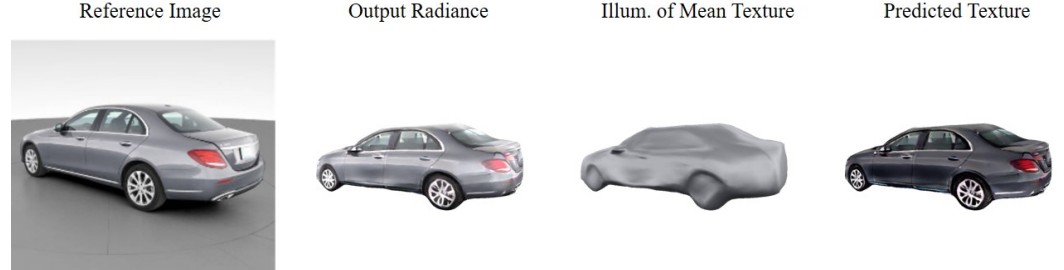

Figure 6: **Gray texture-illumination ambiguity.** For gray cars, there is a brightness ambiguity between texture and lighting. Although the car shown here is silver, another possible interpretation is that the car is dark grey with very bright illumination and specularity.

Kiriakos N Kutulakos and Steven M Seitz. A Theory of Shape by Space Carving. *IJCV*, 38(3): 199–218, 2000.

Ben Mildenhall, Pratul P Srinivasan, Matthew Tancik, Jonathan T Barron, Ravi Ramamoorthi, and Ren Ng. NeRF: Representing Scenes as Neural Radiance Fields for View Synthesis. In *ECCV*, 2020.

Matthew Tancik, Pratul P. Srinivasan, Ben Mildenhall, Sara Fridovich-Keil, Nithin Raghavan, Utkarsh Singhal, Ravi Ramamoorthi, Jonathan T. Barron, and Ren Ng. Fourier Features Let Networks Learn High Frequency Functions in Low Dimensional Domains. In *NeurIPS*, 2020.

Matthew Tancik, Ben Mildenhall, Terrance Wang, Divi Schmidt, Pratul P. Srinivasan, Jonathan T. Barron, and Ren Ng. Learned Initializations for Optimizing Coordinate-Based Neural Representations. In *CVPR*, 2021.

Dmitry Ulyanov, Andrea Vedaldi, and Victor Lempitsky. Instance Normalization: The Missing Ingredient for Fast Stylization. *arXiv:1607.08022*, 2016.

Zirui Wang, Shangzhe Wu, Weidi Xie, Min Chen, and Victor Adrian Prisacariu. NeRF–: Neural Radiance Fields Without Known Camera Parameters. *arXiv:2102.07064*, 2021.

Shangzhe Wu, Ameesh Makadia, Jiajun Wu, Noah Snavely, Richard Tucker, and Angjoo Kanazawa. De-rendering the World's Revolutionary Artefacts. In *CVPR*, 2021.

Yang Xiao, Xuchong Qiu, Pierre-Alain Langlois, Mathieu Aubry, and Renaud Marlet. Pose from Shape: Deep Pose Estimation for Arbitrary 3D Objects. In *BMVC*, 2019.

Lior Yariv, Yoni Kasten, Dror Moran, Meirav Galun, Matan Atzmon, Basri Ronen, and Yaron Lipman. Multiview Neural Surface Reconstruction by Disentangling Geometry and Appearance. In *NeurIPS*, 2020.

| Stage | Optimize | Num. Iters | Loss Weights | | | | | |
|-------|----------|-----------|------|----|---------|-------|-----------|------|
| | | | Mask | DT | Chamfer | Perc. | Perc. mean | Reg. |
| 1 | $\{\Pi_i\}_{i=1}^N$ | 500 | 2 | 20 | 0.02 | 0 | 0 | 0 |
| 2 | Above + $f_{\text{shape}}$ | 500 | 2 | 20 | 0.02 | 0 | 0 | 0.1 |
| 3 | Above + $f_{\text{tex}}$ | 1000 | 2 | 20 | 0.02 | 0.5 | 0 | 0.1 |
| 4 | Above + $f_{\text{env}}, \alpha, k_s$ | 500 | 2 | 20 | 0.02 | 0.5 | 0.15 | 0.1 |

Table 3: **Multi-stage optimization loss weights and parameters.** We employ a 4 stage training process: first optimizing just the cameras before sequentially optimizing the shape parameters, texture parameters, and illumination parameters. In this table, we list the parameters optimized during each stage as well as the number of training iterations and loss weights. All stages use the Adam optimizer (Kingma and Ba, 2015). "Perc. refers" to the perceptual loss on the rendered image with full texture while "Perc. mean" refers to the perceptual loss on the rendered image with mean texture.