# OpenReview forum: "NeRS: Neural Reflectance Surfaces for Sparse-view 3D Reconstruction in the Wild"
_NeurIPS.cc/2021/Conference — NeurIPS 2021 Poster_

### Official Review · Reviewer_MM89 · 2021-07-11

**Rating:** 4
**Confidence:** 4

**Summary:**

This paper presents a method for surface geometry and surface reflectance estimation from multiple photographs in the wild.  The central idea can be viewed as an extension of the NerF-type neural implicit volumetric representation, here the goal is instead to learn neural implicit surface  representation for both object geometry and surface reflectance.

**Ethical Concerns:**

N/A.

**Limitations And Societal Impact:**

Yes.

**Main Review:**

This paper  extends the idea of NeRF from volumetric  model to neural surface modelling.   This idea is very similar to the work of Yariv et al in NeurlPS2020 for disentangled multi-view object geometry  recovery (also represented in the form of neural surface) and object reflectance estimation (also used a similar photometric model).    The overall computation pipeline adopts the inverse graphics rendering procedure where a pre-defined mesh (here a simple mesh) is gradually deformed from a canonical shape (a sphere)  to the target shape to conform with the multi-view image measurements.    Rather than directly predicting 3D vertex positions on the mesh,  the paper propose to predict a displacement field (i.e. mesh flows).

Advantages:
1. impressive experiment results: from a sparse set of multi-view images ( can be as few as 8), the method is able to recovery the shape and texture of objects with general environment lighting. In contrast, NerF often needs more images (>20~40).
 2.  Estimating displacement field rather than shape itself will in principle make the neural net more generalisable to more complex shapes.  ( though no ablation study is provided  in the paper).
3.  Contribute a new  "in the wild" multi-view object dataset (the Craiglist Multi-View dataset).

Limitations:

1. Limited originality:   The originality and novelty of the main idea of the paper is relatively limited.  Apart from some significant overlaps with Yariv NeurlPs 2020,  as well as with Piex2Mesh++ (in CVPR2019),  the idea of estimation mesh deformation field (i.e. Mesh Flow) rather than mesh itself was also proposed in "Kunal Gupta and Manmohan Chandraker, Neurla MeshFlow"  (Neurlps 2020) for the very same task of  image2mesh reconstruction by deforming a canonical initial mesh.    However, no comparison is presented in the paper.   The Pixel2Mesh++ in CVPR2019 also used the same deforming a sphere and rendering and comparing it with the input multi-view images, despite using an explicit surface representation (rather than implicit neural surface representation).   The NeuralMeshFlow pipeline is almost identical to the paper's, which also estimates  a deformation field for neural surface representation.   Doing the above comparisons is imo essential for demonstrating the value of this work.


2. Insufficient validation:    As shown in the figures of the paper,  most of the objects being reconstructed have rather simple  (sometimes convex) geometry shapes  ( e.g. ketchup bottle, a sedan/car,  a rice cooker, a coffee machine,  etc.).   For these simple shapes,  the method of "shape from multi-view silhouettes" may already work well to recover the shape.   Indeed,  I notice the "object mask-loss" used in your method is precisely for this purpose.   Unfortunately, there is no ablation study, nor comparison in your paper to validate this.  It is unclear whether the good result is largely due to the use of the object mask-loss alone.  Which terms plays what role ?     In light of this, the paper is clearly pre-mature, and more work is needed to  validate the main idea.

3. Unclear analysis:  Similarly,  although the paper claims it also estimates a spherical light map (environment map), the efficacy of this env-map has never been demonstrated or validated in the paper.   After all, the Phong shading model is an overly simplified  lighting model not too far away from the dichromatic model.   The recovered texture map shown in the paper (figure-7 in the main paper, as well as in the demo videos) show  strong highlight/specularity-- suggesting  the reflectance estimation is quite off by your method.   Have you evaluated how accurate the recovered env-map, or the recovered Phong parameters ?

More importantly,  it is unclear how this estimated spherical environment map resolves the "fundamental ambiguity" (in reflectance and lighting).  Given the env-map is an area light source,   how can the network distinguish a low-textured object illuminated by a high-frequency env-map, and a highly-textured object illuminated by a smooth env-map ?   To speak of the least,  there is no account nor experiments about how well/accurate the estimated env-map looks like or works.

5.  The paper claims ( in line 120) the method only requires approximately calibrated camera parameters to start with.  However, based on the description in the experiment section (line 236-243), it seems the camera parameters are optimized BEFORE optimizing the other parameters (f_shape and f_env and f_tex), and kept fixed afterward.     This is strange, because if you do not have a good enough initial f_shape, f_texture, and F_env the environment map estimations,   how can you expect to get the camera calibration parameters right or accurately ?    The experiments do not seem to support your claim.

6.  Most of the objects shown in the paper (as well as in the video) have the spherical topology.  While the implicit surface representation naturally accepts non-sphere topologies,  the rasterization graphics rendering step often has issue with surface with self-intersection or holes.   The paper failed to analyze this point.  Tests on more complex shapes  (e.g. with non-spherical topology) is necessary.


Overall,  the paper combines multiple interesting ideas, and experiment results looks good, but its execution (of those ideas) is rather poor,  weak and insufficient.   Its theoretical novelty is limited, and experiment  are also lacking (missing necessary comparisons and ablation studies).   The quality of the paper is okay but not good enough.  It is clearly under-developed and pre-mature, making it just below the bar for NeulPS.




**Time Spent Reviewing:**

6 hours reading + 1 hours writing.

---

> ### Author Response · Authors · 2021-08-10
> **Addressing Positioning, Shape, Illumination, and Initialization**
>
> Originality: Please see the  top-level comment for new evaluations for IDR. We hope these added experiments highlight the importance of our geometry and appearance representations in contrast to the SDF-based geometry and unfactored appearance representations used in IDR.
>
> While we agree that prior approaches, including Neural Mesh Flow and Pixel2Mesh++, have modeled shapes as point-wise deformations from a canonical sphere, **our work tackles a fundamentally different problem setup compared to these**. In particular, both these methods require thousands of ground-truth 3D shapes for learning, whereas our work shows this representation can be inferred using just a single imageset (and no training data).  Moreover, as these methods do not recover  texture or illumination,  it is not possible to evaluate them on the task of novel view synthesis. In addition, Neural Mesh Flow cannot take in multiple viewpoints, and Pixel2Mesh++ assumes known ground truth cameras. In summary, while we agree with the reviewer that prior works have leveraged similar shape representations, our contributions pertain to learning these for sparse view reconstruction in-the-wild, while modeling view-dependent appearance, with noisy cameras and no prior training data.
>
> Validation of geometry from silhouettes: First, we want to note that all our baselines (including NeRF and IDR) make use of mask loss in their optimization, and this clearly highlights that our results cannot be solely attributed to this loss. We agree with the reviewer that it is an important signal, but would emphasize that it is our shape/appearance representation in conjunction with this that lets us recover the final shapes.
>
> To highlight this, we provide several visuals in the rebuttal webpage (https://ners_neurips21.gitlab.io/rebuttal/) for the shape obtained using the standard “shape-from-multiview-silhouettes” technique of visual hulls. We find that the shape model learned by our approach can recover detailed shape information, such as the sideview mirrors of a car or the concavities of an espresso machine, that are not recovered from the silhouettes alone (given the imprecise cameras and limited views).
>
> Illumination and Ambiguities: Given images under a fixed but unknown illumination, it is not possible to factor illumination from material properties (e.g. white car under gray lighting vs gray car with white light, or whether a shiny highlight occurs due to a directional light source vs an exotic material with an uncommon BRDF). Thus, the task of recovering a free-form BRDF and general illumination from a sparse set of views is fundamentally ill-posed.
>
> One option, adopted by most existing single-view works, is to ignore view-dependent effects and learn a textured model, but this does not allow view-dependent effects as we show [here](https://ners_neurips21.gitlab.io/rebuttal/). Another option is to model arbitrary view-dependent effects (e.g. NeRF), but this does not generalize to sparse views in the wild (as we showed in our supplementary results). Ours is an initial attempt toward something in the middle.  We do not wish to claim that ours is the ‘right’ factorization and would be happy to modify any parts of the text that suggest this. Our goal is to produce (one of potentially many) factorizations that let us generate accurate novel views, which can still be quantifiably and rigorously evaluated with novel view synthesis, as we do.
>
> Initialization: The purpose of the approximately calibrated cameras is to serve as an initialization. Camera pose optimization is extremely susceptible to getting stuck in local minima. The reviewer is correct that the camera parameters are optimized (“pretrained”) using the initial f_shape with a mask loss. (See Table 2 in the response to Reviewer h3Zr for an evaluation of pre-training.) However, the camera parameters can still change during the rest of the optimization process. We find that this optimization is still effective even with a coarse (e.g. cuboid) shape initialization. See the rebuttal webpage (https://ners_neurips21.gitlab.io/rebuttal/) for more examples.
>
> Spherical Topology: We agree with the reviewer that this is a limitation of our method. Because the shape model is diffeomorphic to a sphere, we are limited to objects with genus-0 topologies. Nonetheless, we demonstrate that our shape representation can represent a wide variety of object categories.

---

> ### Author Response · Authors · 2021-08-26
> **Final Discussion and Clarifications**
>
> Dear Reviewer MM89,
>
> We hope that you had a chance to read the rebuttal (and also the common reply above) as the discussion period is ending soon. In particular, we hope that the new IDR baseline and ablations address your concerns about validation, and that the clarifications about the task and initialization address your concerns about originality.
>
> Please let us know whether you have any further concerns we could address before the discussion period ends.
>
> Thank you!

---

> > ### Comment · Reviewer_MM89 · 2021-08-26
> > **Still no fully convinced.**
> >
> > Thank authors for the response.  I am still not fully convinced.  The presented test results were mostly on relatively simple (blobby) shapes and the use of object mask obscures the assessment.    I 'd like to keep my original rating, namely, "the work is okay, but not good enough" for Neurlps, based on the paper's current form.    The idea's potential has not been fully demonstrated, but promising.

---

> > > ### Author Response · Authors · 2021-08-27
> > > **Addressing the Contributions and Significance of our Paper**
> > >
> > > Dear Reviewer,
> > >
> > > Thanks for your response. We are glad you found the idea promising, but would like to respectfully disagree with the assessment that its potential has not been well demonstrated.
> > >
> > > First, for a single paper to “fully” demonstrate the potential of an idea (i.e.  explore all/most variants, limitations, extensions etc.) is an unfortunately impossible task - this naturally happens over time as the community adapts, extends, and modifies an idea - and we should not reject a paper simply because the idea has not been perfected. Instead, we argue (and hope the reviewer agrees) that the goal of a paper should be to “convincingly” demonstrate the merits of an idea, by showing clear benefits over alternatives in practically relevant setups - and that is exactly what we do!
> > >
> > > We would request the reviewer to ask themselves two simple questions:
> > >
> > > 1. Does the paper demonstrate clear and significant improvements over baselines, and in a fair empirical setup?
> > > 2. Is this setup representative of an important and practically relevant problem?
> > >
> > > We believe the answer to both the questions above is an unequivocal ‘yes’, and argue that this constitutes a clear and convincing demonstration of the merits of our idea - one we think the community would benefit from.
> > >
> > > We also would like to re-address the two specific concerns raised:
> > >
> > > 1. ​​”use of object mask obscures the assessment”: We feel this is a slightly unfair critique, and one that we extensively addressed in our previous response. In particular:
> > >     * First and foremost, all our baselines similarly use object masks! Thus it is not “object-mask alone” that leads to our improved results, but rather our method/representation in conjunction with the available learning signals.
> > >     * We added visualizations to compare our inferred shapes to “shape-from-multiview-silhouettes” methods to further alleviate this concern.
> > > 2. Results on “relatively simple shapes”: While we do show results on more arguably more complex objects e.g. espresso machine with concavities, and controller with protruding joysticks, we definitely agree with the reviewer’s sentiment that our approach has limitations in terms of the shapes it can represent (e.g. only genus-0).
> > >
> > > But should this really be a reason for rejecting our work? We have demonstrated that, despite its limitations, our approach can represent several everyday objects and in settings that baselines cannot handle well. Thus even in its current form, our approach is of clear practical relevance, and presents key ideas that the community could build upon.
> > >
> > > Thank you,
> > >
> > > Authors

---

### Official Review · Reviewer_3udv · 2021-07-15

**Rating:** 6
**Confidence:** 3

**Summary:**

This work introduces a method to reconstruct a scene from a small number of views. Rather than using a volumetric representation, it uses an explicit surface that is defined as a sphere with perturbed vertices. This restriction allows the model to learn to decompose the reflectance into a Phong lighting model. A contribution of the work is the introduction of a large, in-the-wild multiview dataset of cars from craigslist.

**Limitations And Societal Impact:**

I believe that few view 3D reconstruction has the ability to give non-professional users the ability to create 3D assets. Negative social impact is adequately discussed.

**Main Review:**

Pros:

+ Surface representations deserve more exploration, and using the additional constraints for few view reconstruction makes sense.

+ Real world dataset with few views is a great contribution and space to work in. I really like the idea of scraping marketplaces for such data, and this work introduces a new dataset consisting of millions of multi-view captures, which is great. The main downside is that as far as I can tell, they are only for cars.

+ BRDF factorization from sparse views is very hard.

+ Can work with only coarse cameras (by jointly optimizing camera parameters during reconstruction).

+ Formulation is clean and elegant, everything defined on the surface of a sphere (geometry, env map, texture).

+ Can use better reconstruction losses due to entire images generated at train time.

Cons:

- My biggest concern is that this approach is missing comparisons to other surface based models! for example: [Yariv et al] with code available, why was this not compared to? This would be key to showing that the "deformed sphere" approach is better than an SDF representation, as it comes with some limitations in terms of what kinds of models you can represent.

- Results are only okay, they tend to look like "balloon" versions of the original, and the Phong shading model does not capture specular reflections well (at all?).

- Requires a mask loss (like Yariv et al), but it seems like the mask loss might already get you a decent starting model simply by reconstructing the visual hull from the masks for convex objects like cars. Is it possible to compare the result to this simple solution?

- Requires per-category initialization mesh. In fact, this part was a bit unclear to me, what do you use for cars? How do you show motorcycle results? Where do the other non-car examples and initialization meshes come from (e.g., the coffee machine)? What is the importance of the initialization mesh? Is it just for camera pose estimation, or do you use it to initialize the geometry network as well?

Additional Questions:

When comparing to baselines, the final camera positions that are derived by optimizing this method's objective are used. In some senses, those may be optimal for this approach. How much does the camera change 1) between initialization and the pre-training, 2) between pre-training and final training?

It is interesting that Nerf outperforms with respect to LPIPS but seems clearly worse in the results. Do you have any intuition why this is?

How do you model shadows? This seems important given that all images are outdoor, and you often see the object shadowing itself. Does the neural renderer correctly model these?

TL/DR: I like this work, but I think its missing some comparisons (e.g., to another surface based method), and the overhead of category specific steps may make it hard to extend to new classes in the wild.

**Time Spent Reviewing:**

4

---

> ### Author Response · Authors · 2021-08-10
> **Addressing Shape Initialization and Optimized Cameras for Baselines**
>
> Comparison: Please see the top-level comment for the IDR baseline.
>
> Result quality: We wish to place our results in context. The impressive results that the community has seen over the last two years using NeRF/IDR and related approaches are in settings with hundreds of images of a single object and/or precisely known cameras. While we agree that our reconstructions are not of a similar quality, we feel that given the challenging problem setup, it is perhaps unfair to expect them to be! In fact, prior to our work, it was an open question whether any reconstruction and appearance modeling approach could even leverage such limited image data with coarse viewpoints, and we would argue that our results are impressive when put in this context.
>
> Mask loss: We agree that the mask loss is essential for learning accurate shapes (for NeRS and baselines). However, just using foreground masks via volume carving is not sufficient by itself to yield accurate shapes—and we find that our surface representation and joint optimization is important. As volume carving does not allow camera optimization, we visualize the visual hull reconstruction results using cameras from different stages in the [rebuttal webpage](https://ners_neurips21.gitlab.io/rebuttal/).  We find that the shape learned through our joint optimization correctly captures details that cannot be learned from the silhouettes alone, such as the shape of the sideview mirrors.
>
> Initialization: Our approach is robust to the initialization being coarse. All car models are initialized from the same sedan initial mesh. The motorcycle was initialized from an existing 3D motorbike mesh. We demonstrate that all categories other than cars and motorcycles can be initialized from an axis-aligned cuboid with just 3 degrees of freedom. We re-ran the coffee machine with a cuboid initialization as well. Please see [rebuttal webpage](https://ners_neurips21.gitlab.io/rebuttal/) to see mesh initializations. The initial mesh is used to both pre-train the camera poses and initialize the geometry network.
>
> Additional clarifications:
>
> **Do optimized cameras hurt baselines?** We would in fact argue the opposite! While NeRS is tasked with jointly optimizing the initial coarse cameras along with shape and appearance, all our baselines get the added benefit of already optimized accurate cameras. If anything, this biases the evaluation in the favor of the baselines as they do not need to jointly optimize cameras.
>
> To illustrate this, we visualize NeRF models trained without these optimized cameras in the rebuttal webpage (https://ners_neurips21.gitlab.io/rebuttal/). We clearly see that giving this baseline our optimized cameras significantly improves results. A similar trend is also seen with our IDR results, where allowing IDR to further optimize cameras hurts performance. We therefore argue that allowing NeRF/IDR based methods to further optimize cameras in the sparse-view setting leads to (even) more overfitting and further degrades their performance, and providing them with our optimized cameras helps give an upper-bound on their performance.
>
> * We would like to clarify that NeRS outperforms NeRF in LPIPS (lower is better) as shown in Table 1 in the main paper.
> * We do not explicitly model shadows or self-occlusions, but the texture and illumination map can model these types of lighting effects by “baking” them into the respective components. Refer to the [rebuttal webpage](https://ners_neurips21.gitlab.io/rebuttal/) for a visualization.

---

> ### Author Response · Authors · 2021-08-19
> **Happy to Provide Additional Clarifications**
>
> Dear Reviewer 3udv,
>
> You mentioned that you “like the work” but that the “biggest concern is that this approach is missing comparisons to other surface based models”. In our response, we added quantitative and qualitative comparisons to IDR (in addition to also addressing other concerns).
>
> We were wondering if you would kindly re-evaluate your rating in light of this—please let us know if any more details or clarifications would help.
>
>
> Thanks,
>
> Authors

---

> ### Author Response · Authors · 2021-08-26
> **Final Discussion and Clarifications**
>
> Dear Reviewer 3udv,
>
> We hope that you had a chance to read the rebuttal (and also the common reply above) as the discussion period is ending soon. In particular, we hope that the new IDR baseline addresses your primary concern about the missing comparisons with surface-based models, and that we sufficiently addressed your concerns about the mask loss, initialization, and cameras.
>
> Please let us know whether you have any further concerns we could address before the discussion period ends.
>
> Thank you!

---

### Official Review · Reviewer_h3Zr · 2021-07-16

**Rating:** 6
**Confidence:** 4

**Summary:**

This paper tackles the problem of recovering 3D shape, surface albedo, specular material (Phong model), and environment lighting map from a sparse set of images from ~8 viewpoints, given a rough template shape as initialization and automatically obtained masks and camera poses. This is achieved by coupling a AtlasNet-like representation for shape and texture with explicit material and lighting models in an optimization pipeline that reconstructs the input observations.

The paper shows plausible decomposition results on multi-view image sets from the Internet with rough camera estimates and mask predictions, whereas existing NeRF-based methods fail dramatically on such few inputs.


**Ethical Concerns:**


I do not see immediate concerns.


**Limitations And Societal Impact:**


A few limitations are discussed, and visual examples are provided in the supplementary material. It would also be interesting to show how the method fails without sufficiently accurate initial mesh.


**Main Review:**


## Strengths
### S1 - Method
- Overall, the proposed method is carefully designed and achieves compelling decomposition results.
- It uses an AtlasNet like representation for 3D shape, texture and surface material, which is discretized and rendered into 2D images through a differentiable renderer. In this view, the proposed method essentially extends an AtlasNet-based multi-view reconstruction model with material and lighting components.
- The environment lighting map is modeled similarly with a sphere AtlasNet, and sampled for rendering.

### S2 - Results
- The decomposition results are visually appealing, and no existing methods seem to be able to achieve similar results on real world photo sets "in the wild". The comparisons against NeRF-based methods also show significant performance gap.
- The ablation results with naive NeRF-like view-conditioning texture prediction reported in the supplementary material are very insightful, confirming the effectiveness of enforcing explicit shading models for few-view reconstruction.

## Weaknesses
### W1 - Initialization
One significant limitation of the proposed method is need of a category shape template for shape initialization. It also relies on another method for estimating coarse initial camera poses. These additional information has significantly simplified the problem.

### W2 - Over-claim on shape representation
The shape representation they are using is essentially an AtlasNet over a sphere. I think its novelty has been over-claimed by focusing the comparison instead on NeRF and its rendering procedure.

### W3 - Simplistic material model
- The method assumes a Phong model with a global shininess to model specularity, which cannot model sophisticated real world illumination effects, resulting from complex spatially-varying BRDF.
- It also assumes a gray-scale environment map, which is not realistic.
- It seems such explicit material and lighting models, especially with the unrealistic Phong model, also limit the image quality, resulting in relatively low faithfulness scores in novel view synthesis as shown in Table 1.

### W4 - Limited comparison
- The method is only compared to NeRF-based models. How does the method compare to IDR and other photometric stereo methods? Can this CVPR20 work [A] handle these cases, which also recovers 3D shape and SVBRDF from a small set of images, but seems to assume collocated flash light?

[A] "Deep 3D Capture: Geometry and Reflectance from Sparse Multi-View Images". Sai Bi, Zexiang Xu, Kalyan Sunkavalli, David Kriegman, and Ravi Ramamoorthi. CVPR 2020.

## Additional comments
- How is the shape network $f_\text{shape}$ pretrained with the template mesh? I suppose the mapping from sphere to surface is unknown. Is this achieved using for example an occupancy loss?
- How well is the environment lighting estimated? There is only one single example in Fig 4. It would be helpful to show more visualizations.
- What are the resolutions of the discretized mesh and environment map?


**Time Spent Reviewing:**

4

---

> ### Author Response · Authors · 2021-08-10
> **Clarifications on Initialization, Shape, and Materials**
>
> Initialization: We respectfully disagree with the reviewer that the shape and camera initialization is a significant limitation. For many objects, including new results on the Espresso machine, a cuboid is sufficient as a category shape template (See linked page: https://ners_neurips21.gitlab.io/rebuttal). While we require a few (8-16) coarse initial poses, existing approaches require many (50+) precise camera poses.
>
> Over-claiming on shape representation: We agree that our shape representation is similar to AtlasNet and IMR, and we acknowledge this in the text (L140). The key contributions in our approach lies in learning these parameters while also learning view-dependent effects from limited views. We will modify the text to make this more clear.
>
> Simplistic Material Model: There is a natural spectrum of possible appearance models. On the one extreme, several prior approaches (e.g. IMR) only represent textured surfaces and cannot model any view-dependent effects. On the other extreme, allowing arbitrary view-dependence (or equivalently arbitrary BRDFs) e.g. in NeRF offers a very flexible representation, but is not a robust solution for sparse-view settings (please see “NeRS + Naive View dependence” experiments in the supplementary). While we agree with the reviewer that our simplified material model has limitations, we argue it offers an empirically robust middle-ground between the two extremes described above, and we hope future attempts will further explore this spectrum.
>
> Limited Comparison: Please see top-level comment for empirical and qualitative comparison with IDR. While Deep 3D Capture [A] also targets the few-view setup, it requires carefully captured images with known camera poses and point lights to be collocated with the cameras. We assume camera poses are noisy and make no assumptions about the direction of environment lighting.
>
> Additional Comments:
> * We use an [off-the-shelf approach](https://github.com/icemiliang/spherical_harmonic_maps) to learn a sphere-to-mesh correspondence. Then we apply standard chamfer and shape regularization losses to train $f_\text{shape}$.
> * We show some qualitative examples of the environment lighting in the linked webpage here: https://ners_neurips21.gitlab.io/rebuttal.
> * We discretize the mesh with 4002 vertices and the environment map with 1002 rays.

---

> ### Author Response · Authors · 2021-08-26
> **Final Discussion and Clarifications**
>
> Dear Reviewer h3Zr,
>
> We hope that you had a chance to read the rebuttal (and also the common reply above) as the discussion period is ending soon. In particular, we hope we hope that our clarifications about the initialization and materials model addressed your concerns, and the new IDR baseline further strengthened our empirical evaluations.
>
> Please let us know whether you have any further concerns we could address before the discussion period ends.
>
> Thank you!

---

> ### Comment · Reviewer_h3Zr · 2021-09-03
> **Remaining borderline accept**
>
> I appreciate the authors' effort in providing the detailed responses. My rating remains borderline accept. The additional comparison against IDR shows significantly better shape and texture reconstruction. Other additional results are also very insightful.
>
> The novelty of the technical components is quite limited, as the core of the method is still multi-view reconstruction with an implicit surface representation (as in AtlasNet and IMR), but extended to also recover shininess material and (grayscale) environment map.
> Then I think the two key questions are:
> - (a) are the recovered material and illumination properties interesting enough? -- I think they are not great but ok given sparse input views, judging from decomposed results in the rebuttal (Environment Map Visualizations).
> - (b) does this additional appearance modeling help with the shape recovery? -- I think the answer is no, judging from the ablation study in the rebuttal (No Illumination Ablation).
>
> Both of the Phong BRDF and environment map are also rather limited in practice.
>
> It requires a rough initial shape and pose estimates to begin with. While the authors argue that the initialization can be as simple as a cuboid, this is not the case for many examples shown, and I assume it does not work without the initialization. Even the scale and the orientation will give a lot of hint about the shape already.
>
> Overall, despite the limitations of the current method, the results are still interesting enough for this challenging task.

---

### Official Review · Reviewer_kaid · 2021-07-18

**Rating:** 5
**Confidence:** 4

**Summary:**

The submission studies the problem of 3D geometry and appearance reconstruction from a sparse set of 2D images, and proposes a new neural implicit model called Neural Reflectance Surfaces (NeRS). The main difference from prior work (specifically, NeRF) is the following:
1. using a much sparser set of 2D observations as input - typically around 10 images per instance
2. a neural surface model for geometry instead of neural volumetric representations,
3. a neural environment map to model lightings, conditioned only on the viewing direction,
4. BRDFs representation for surface specularity to model view-dependent effects.

In particular, the shape geometry is modeled as a continuous and implicit deformation of a unit sphere, parametrized by a multi-layer perceptron (MLP). Similarly, the diffuse color is parametrized by a second MLP with spherical coordinates as input. To render an image, a continuous unit sphere in 3D is first discretized into a set of vertices (spherical coordinates), which are then passed as input to the shape MLP to construct a triangulated surface mesh. The per-vertex diffuse colors are computed from a second MLP using the same set of spherical coordinates. Together with the neural environment map (from a third MLP), the final pixel intensity can be rendered using a Phong reflection model. These 3 different MLPs are optimized sequentially and iteratively using a combination of 2D image loss, a mask loss, and mesh regularization. The method is compared against different variants of NeRF on a new "Craigslist Multiview" (CMV) dataset. Both quantitative and qualitative comparisons show that the proposed NeRS method outperforms NeRF under this few-shot experiment setup.

**Limitations And Societal Impact:**

Yes

**Main Review:**

#### Originality
- This submission is slightly different from NeRF in the problem setup - it focuses on the few-shot scenario, and reconstructs only a single instance of objects.
- Some of the main technical components are from Tulsiani et al. (2020), including the mesh and color parametrization and the loss function. There is a high level of similarity between the two methods, with some differences in using environmental map and BRDFs as well as the number of inputs.
- Related Work:
    - Please elaborate on the differences from Tulsiani et al. (2020).
    - The idea of using surface representation is relevant to "Deferred neural rendering: Image synthesis using neural textures" (ToG, 2019). Please consider adding this (and its follow-up works) to the related work section.
    - Some papers on modeling reflection in the context of NeRF is also relevant, including but not limited to "NeRV: Neural Reflectance and Visibility Fields for Relighting and View Synthesis" and "Neural Light Transport for Relighting and View Synthesis"

#### Quality
- The submission is technically sound.
- Both the quantitative and qualitative results show that the proposed method significantly outperforms NeRF in the sparse-input regime on the new dataset.
- However, the experiment section is limited in several major ways:
    - The submission is only tested on one new dataset - it would be much more convincing if the authors can include experiments on ShapeNet and/or other datasets that are widely used in the community.
    - There are a number of variables in the optimization problem, including geometry, textures, camera poses, and specularity. An ablation study on the contribution of these optimization variables to the final accuracy is needed - for instance, how important is it to include the camera poses and the specularity in this optimization framework, or can we remove them without hurting the accuracy? Showing the quality metrics in Table 2 (in the supplementary materials) after each optimization stage would be helpful.
    - It is unclear why the environment map requires a MLP representation. Can we achieve the same performance with bilinear sampling from an optimizable tensor? Please add an ablation study to support your design choice.
    - Another great ablation study to include is a curve of reconstruction quality vs. number of input images. This helps readers understand how many images are needed for their use cases.
- While this submission is interesting to the community, it seems like a work-in-progress rather than a complete piece of work, especially given the limited experiments.
- Instead of directly claiming NeRF fails, the authors have made an effort to improve NeRF with several modifications to ensure a fair comparison. The extensive visualization provided by the authors is quite helpful.

#### Clarity
- The paper is overall clearly written and well organized, and the supplementary materials are also helpful.
- I believe it is largely reproducible based on the technical description.
- A few comments that might help improve the writing:
    - Please consider adding new paragraph in Section I as a summary of the main contributions, as well as the differences from NeRF and Tulsiani et al. (2020).
    - It is not crystal clear from the paper whether the specular material parameters $(k_s, \alpha)$ are defined on an instance level (e.g., shared by the entire surface of an object instance) or on a per-pixel level (parametrized by another MLP). My understanding is the former, but an explicit explanation would help.
    - One missing implementation detail is how the authors sample the view directions $\omega \in \Omega$.

#### Significance
- The results are significant, since it might help enable 3D reconstruction from a sparse set of 2D images. I believe the topic is of great relevance to the machine learning / computer vision / computer graphics community and beyond.
- I can see practitioners using this technique to build new applications that require less intensive data capturing from users. Researchers could also build on top of the submission in terms of complexity, extending from single objects to complete scenes.
- Compared to NeRF, the submission's task is more challenging in the input size but at the same time simpler in scene complexity (single object vs. entire scene). It is addressing a simpler task than Tulsiani et al. (2020) given that it is using more input images.
- It improves the state of the art on few-shot reconstruction in a clear and demonstrable way.
- It also provides a new, unique dataset for few-shot reconstruction, establishing a new benchmark for future exploration.

**Time Spent Reviewing:**

5

---

> ### Author Response · Authors · 2021-08-10
> **Additional Ablations and Clarifications**
>
> Originality
>
> * Related work: We thank the reviewer for the pointers to the additional papers exploring surface-based representations. We will include these references in the manuscript. With respect to Tulsiani et al (Arxiv 2020), we note that the primary similarity is the deformed sphere-based representation of shape. The texture representation is also similar although Tulsiani et al predict texture flow rather than RGB directly as we do. The key difference is that we target an entirely different setup; IMR is a single-view object-specific approach trained on category-level supervision whereas we propose a multi-view instance-specific approach that does not rely on a training dataset, and can also model view-dependent effects. To be clear, the only images ever seen by NERS are the ~10 multiview images of a single target object to be reconstructed, but IMR requires thousands of images of many instances of a given category. Because NERS ultimately relies on test-time optimization rather than a training set, it may generalize better because it is less susceptible to mismatches between train and test, and would be directly applicable to generic objects (e.g. ketchup bottles) unlike IMR which would require thousands of new category-level training images.
>
> Quality
> * Test dataset: We evaluated our approach (and baselines) on the CMV dataset because it is representative of the challenging real-world setup our approach aims to solve: a) natural lighting conditions, b) varying texture and materials, c) imprecise camera poses with realistic errors (instead of gaussian noise added to calibrated ones). Unfortunately, ShapeNet and other synthetic/controlled capture datasets simplify several of these challenges (e.g. only have simplistic lighting) and were not ideal as testbeds. We would like to point out that our qualitative results using custom captures of everyday objects show that our approach works on generic objects beyond those in the CMV dataset.
>
> * Optimization variables: In the full optimization, optimizing all variables is critical. For instance, the shape initialization is generally dissimilar from the actual geometry, so optimizing the instance-specific shape model is necessary for a reasonable reconstruction (e.g. the shape changes from a sedan to a jeep, or from a cuboid to an espresso machine). The primary purpose of the stage-wise training is to pre-train components of the full model, in particular the camera poses and the shape model. To evaluate the importance of this pre-training, we ran a new ablation experiment in which we skip this pre-training (see Table 2 below). Note that we ran these ablations on the same evaluation subset as the IDR evaluation.
>
> **Table 2: Pre-training ablations**
>
> | Ablation | PSNR $\uparrow$  | SSIM $\uparrow$  | LPIPS $\downarrow$    | MSE  $\downarrow$|
> | - | - | - | -| - |
> | NeRS (Ours)    |     **15.1** | **0.666** | **0.204** | **0.0356** |
> | NeRS w/ no Camera Pre-training  |    10.9  |0.569 | 0.407| 0.0887|
> | NeRS w/ no Camera + Shape Pre-training   |  10.8 | 0.559 | 0.427|  0.091  |
>
> We also include a qualitative result without modeling the view-dependent effects in the [supplemental link](https://ners_neurips21.gitlab.io/rebuttal/).
>
> * Reconstruction quality vs number of input images: We agree with the reviewer that analysis of the relationship between the number of images and view synthesis quality would be informative for readers. We ran an initial experiment evaluating view synthesis quality for a specific instance which suggests that performance begins to saturate after 11-12 images, shown in Table 3. We will include more detailed analysis in the paper.
>
> **Table 3: Number of Images Experiment**
>
> |  |  |  |  |  |  |  |  |  |  |  |  |  |  |  |
> |- |- |- |- |- |- |- |- |- |- |- |- |- |- |- |
> | Number of Images | 1 | 2 | 3 | 4 | 5 | 6 | 7 | 8 | 9 | 10 | 11 | 12 | 13 | 14 |
> | LPIPS ($\downarrow$) | 0.33 | 0.29 | 0.38 | 0.19 | 0.18 | 0.17 | 0.18 | 0.16 | 0.16 | 0.16 | 0.15 | 0.15 | 0.14 | 0.15 |
>
> * Implicit environment map: We agree with the reviewer that both choices (modeling the environment map as an implicit function or a directly optimizable tensor) might suffice. We made the design decision to use an MLP for primarily aesthetic reasons—it allowed for a unified representation across shape, texture, and illumination (all modeled as mappings for points on a sphere). An image-based environment map (similar to Wu et al 2021) would likely work, although we conjecture that an MLP representation which allows for arbitrary resolution may scale better.
>
> Clarifications: The specular parameters $k_s$ and $\alpha$ are indeed defined at an instance level. To compute the radiance integral over the hemisphere, we sample rays on a unit sphere and ignore any rays pointing away from the normal. We will include these details in the paper.

---

> > ### Comment · Reviewer_kaid · 2021-08-27
> > **Keeping original ratings**
> >
> > I would like to thank the authors for their explanations and additional experiments.
> >
> > However, my rating remains 5 (Marginally below the acceptance threshold). My main concerns were limited novelty (core model highly similar to Tulsiani et al.), lack of ablation studies, and the choice of using only one dataset. I was not convinced by the authors' arguments that test-time optimization as a problem setup is novel, nor that the CMV dataset is the only viable choice. The use of MLP for the environment map also seems to be an overkill.
> >
> > With that said, the new results on accuracy vs. #input images would be great to have in future updated versions.

---

> ### Author Response · Authors · 2021-08-26
> **Final Discussion and Clarifications**
>
> Dear Reviewer kaid,
>
> We hope that you had a chance to read the rebuttal (and also the common reply above) as the discussion period is ending soon. In particular, we hope that the ablations and number of views experiment address your concerns about evaluation, and that our responses highlight how the setup our approach tackles is uniquely different from prior work (e.g. we model view-dependent appearance and require no training dataset unlike Tulsiani et. al.).
>
> Please let us know whether you have any further concerns we could address before the discussion period ends.
>
> Thank you!

---

### Author Response · Authors · 2021-08-10
**Evaluation of IDR and Clarification of Initialization**

We would like to thank the reviewers for their detailed feedback. Before we address reviewer concerns, we want to highlight to the AC and the reviewers why we believe this work is particularly exciting.

Our approach allows reconstructing a generic object while modeling its view-dependent appearance, and it can do this using just a small number of natural images under realistic lighting. This is a far more challenging setting than tackled by **any** recent or classical method, all of which assume at least one (and often more) among: i) an order of magnitude more images for the test object, ii) precisely known accurate camera poses, or iii) a large training dataset for learning. Our approach presents the first results for reconstruction and view synthesis without any of these assumptions, and we would argue that this is a strong empirical result which would be of a broad interest. We certainly agree that there are limitations to our approach, but feel that this work is still an important initial step. In particular, our core contributions pertaining to neural surface representations with factorized view-dependent appearance modeling should pave the way for future extensions which handle these limitations.

We address the specific reviewer concerns in individual responses but would like to first address common requests for comparisons to SDF-based (instead of NeRF-like volumetric) implicit representations. We thank the reviewers for this suggestion, and note that even these methods have only been validated in more curated settings, requiring orders of magnitude more images and precise calibrated cameras. To empirically validate their performance in our setting, we used IDR (Yariv et. al.) on a representative subset of our evaluation dataset and measured its view-synthesis performance against NeRS. We will update the main paper with evaluation over the entire evaluation set. We evaluated IDR with fixed and trainable camera parameters, both initialized from the optimized cameras from our method (See Table 1). We observe that NeRS significantly outperformed IDR across metrics and include qualitative results at this anonymized webpage (https://ners_neurips21.gitlab.io/rebuttal/). While both NeRS and IDR use similar supervisory signals (mask and image reconstruction objectives), we find that our surface-based representation acts as an effective regularizer for sparse-view reconstruction. We find that IDR outputs reconstructions with “bubbly” surfaces when insufficient views, suggesting that SDF-based geometry may be underconstrained. We hope this additional experiment assuages one of the primary concerns shared by the reviewers, and further highlights that no prior approach is well-suited to the challenging setup our work addresses.

**Table 1: IDR results**

| Method    |  PSNR $\uparrow$  | SSIM $\uparrow$  | LPIPS $\downarrow$    | MSE  $\downarrow$|
| - | - | - | - | - |
| NeRF        |      **16.0**  | **0.707** |  0.284 | 0.0456 |
| IDR (Trained Cameras)  |            11.8 |  0.597 |  0.383 |  0.094 |
| IDR (Fixed Cameras) | 13.4 |   0.646  |  0.334 |  0.078 |
| NeRS (Ours) | 15.1 |  0.665 |  **0.204** |  **0.0356** |


We address another common request to clarify our assumptions on initialization; particularly, initialization of cameras and object shape. For our quantitative benchmark on marketplace cars, we make use of imprecise cameras provided by off-the-shelf methods rather than injecting gaussian noise into ground-truth cameras. We see this as a necessary component of any truly “in-the-wild” method. For objects in the wild, we find that assigning cameras to bins of roughly 45 degree increments is sufficient. In terms of object shape, we require only coarse estimates of shape, such as a cuboidal approximation of an espresso machine or bottle of ketchup. Empirically, we find that a human user can roughly bin the cameras and define an HxWxD cuboid in minutes, implying that NeRS can be used to build models of never-before-seen objects in less than an hour on a standard GPU workstation.

To show requested visualizations of IDR, shape initializations, volumetric mask carving, and environment maps, please see this anonymous webpage: https://ners_neurips21.gitlab.io/rebuttal/

---

### Decision · Program_Chairs · 2021-09-27

**Decision:**

Accept (Poster)

**Comment:**

This paper received borderline reject initial reviews, with the reviewers asking for comparisons with IDR etc. and more discussion on the environment maps. The authors provided significant new results which addressed most of the reviewers' concerns. Limitations remain both on the  accuracy of the illumination/reflectance decomposition and on the range of shapes that the method handles.
However, based on the current interest in the area, the novel experimental set-up and dataset and the fact that the authors have demonstrated better results in the case of a challenging sparse view set-up, the metareviewer recommends acceptance. The authors are urged to include the comparisons from the rebuttal to the final paper.